# SaMer: A Scenario-aware Multi-dimensional Evaluator for Large Language Models

**Kehua Feng**[1,2*]**, Keyan Ding**[2*]**, Jing Yu**[2,3]**, Yiwen Qu**[5]**, Zhiwen Chen**[5]**, Chengfei Lv**[5]**,**
**Gang Yu**[5]**, Qiang Zhang**[2,4†]**, Huajun Chen**[1,2†]

[1]College of Computer Science and Technology, Zhejiang University
[2]ZJU-Hangzhou Global Scientific and Technological Innovation Center, Zhejiang University
[3]Polytechnic Institute, Zhejiang University
[4]ZJU-UIUC Institute, Zhejiang University
[5]Alibaba Group
{kehuafeng, dingkeyan, qiang.zhang.cs, huajunsir}@zju.edu.cn

## Abstract

Evaluating the response quality of large language models (LLMs) for open-ended questions poses a significant challenge, especially given the subjectivity and multi-dimensionality of "quality" in natural language generation. Existing LLM evaluators often neglect that different scenarios require distinct evaluation criteria. In this work, we propose SaMer, a scenario-aware multi-dimensional evaluator designed to provide both overall and fine-grained assessments of LLM-generated responses. Unlike fixed-dimension evaluation approaches, SaMer adapts to different scenarios by automatically identifying and prioritizing relevant evaluation dimensions tailored to the given query. To achieve this, we construct a large-scale fine-grained preference dataset spanning multiple real-world scenarios, each with distinct evaluation dimensions. We then leverage a text embedding model combined with three specialized heads to predict the appropriate evaluation dimensions and corresponding scores, as well as the respective weights that contribute to the overall score. The resulting model offers fine-grained and interpretable evaluations and shows robust adaptability across diverse scenarios. Extensive experiments on eight single rating and pairwise comparison datasets demonstrate that SaMer outperforms existing baselines in a variety of evaluation tasks, showcasing its robustness, versatility, and generalizability.

## 1 Introduction

The rapid development of large language models (LLMs) has significantly enhanced their ability to generate human-like responses to open-ended questions. However, assessing the quality of these responses automatically remains a critical challenge in the field of natural language processing. The complexity arises from the inherent subjectivity of "quality" in text generation, which often depends on multiple factors such as relevance, coherence, factual accuracy, and fluency. Conventional automated evaluation metrics (e.g., BLEU (Papineni et al., 2002) and ROUGE (Lin, 2004)) often fail to account for human perception of natural language in terms of its flexibility and complexity in conveying rich yet equivalent semantic information (Zheng et al., 2023b).

In recent years, the utilization of LLMs as judges (Zheng et al., 2023b) has gained significant attention, with cutting-edge LLMs being employed as generative evaluators to provide judgments on text quality, including single rating and pairwise comparison. Previous investigations (Zhou et al., 2023; Fu et al., 2023; Zheng et al., 2023b; Gilardi et al., 2023) suggest that proprietary LLMs can effectively mimic expert evaluations when appropriately prompted; however, they are costly and pose reproducibility issues. As an alternative, fine-tuning an open-source LLM to create specialized evaluators offers a cost-effective approach (Wang et al., 2023b), which has even outperformed proprietary LLMs in certain evaluation tasks. Nevertheless, their flexibility, robustness, and versatility

---

*Equal contribution.
†Corresponding authors.

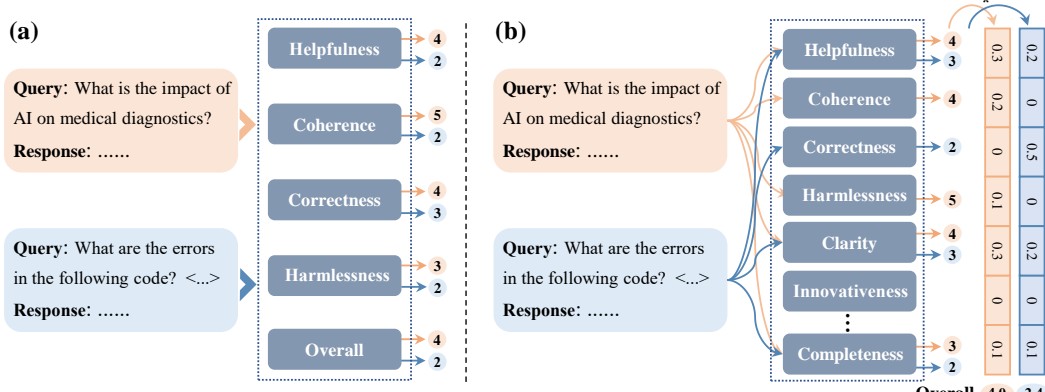

Figure 1: Comparison between **(a)** the conventional multi-dimensional evaluation method and **(b)** the proposed scenario-aware multi-dimensional evaluation approach. The conventional method uses fixed evaluation dimensions for all queries, assigning scores to attributes like helpfulness, coherence, correctness, and harmlessness, which may not reflect the context-specific needs of each query. In contrast, our scenario-aware approach dynamically identifies relevant evaluation dimensions based on the query context. It then assigns scores and weights to each dimension according to its importance for the scenario, allowing for a more adaptive and context-sensitive overall assessment of the LLM's response quality.

still lag behind, limiting their ability to handle diverse real-world scenarios. Moreover, LLM as a judge often exhibits cognitive biases towards aspects such as response length, option order, and entity preference (Huang et al., 2024; Park et al., 2024).

On the other hand, most LLM-based evaluators (Wang et al., 2023b; Li et al., 2023; Kim et al., 2023; Wang et al., 2023a; Kim et al., 2024) only provide an overall score or ranking, even if multi-dimensional evaluations are provided, they are limited to a few coarse-grained and fixed dimensions such as harmlessness, honesty, and helpfulness. *These approaches neglect the need for adaptive fine-grained evaluations, especially when dealing with different types of open-ended questions that require varying dimensions for quality assessment.* For example, the assessment of creative writing should focus on originality, innovation, and coherence, while technical questions might need evaluations around practicality, accuracy, and clarity. This gap calls for a more nuanced evaluation framework that can identify and prioritize the appropriate dimensions for each query and assess them accordingly, providing a comprehensive and interpretable evaluation of LLM responses across diverse scenarios.

To address these challenges, we propose **SaMer**, a scenario-aware multi-dimensional evaluator that not only quantifies overall quality but also provides fine-grained evaluation of LLM-generated responses across diverse scenarios. In contrast to existing approaches (Wang et al., 2024a; Wettig et al., 2024) that rely on a limited set of predetermined evaluation dimensions for all queries, our approach employs a flexible framework tailored to accommodate a variety of assessment criteria and adaptively identifies the appropriate evaluation dimensions based on different question types (as illustrated in Figure 1). Our approach begins by constructing a large-scale pairwise preference dataset covering dozens of scenarios, each associated with distinct evaluation dimensions. We then integrate a text embedding model with three specialized heads: one for predicting which dimensions are required for the given query (i.e., the dimension *predictor*), another for scoring the response quality of these dimensions (i.e., the dimension *scorer*), and the third for weighting the contribution of these dimensions to the overall score (i.e., the dimension *weighter*). These components are jointly trained using a combination of multi-label classification loss and pairwise rank loss to ensure that the model not only learns to discern relevant dimensions but also effectively ranks responses based on their quality in those dimensions. The resulting model shows the following advantages:

- **Flexibility in Evaluation Mode:** SaMer provides overall and fine-grained assessments of LLM-generated responses, supporting single rating and pairwise comparison.

- **Adaptability across Diverse Scenarios:** SaMer dynamically identifies the necessary evaluation dimensions according to different question types, allowing for flexible adaptation to diverse evaluation scenarios.
- **Fine-grained and Interpretable Evaluations:** SaMer provides detailed weights and scores for all dimensions, enabling us to discern the dominant factors in the evaluation and understand how each dimension contributes to the overall quality.

To validate the effectiveness of our model, we conduct extensive experiments on three single rating benchmarks and five pairwise comparison benchmarks. Our results demonstrate that SaMer achieves robust performance and strong generalizability, outperforming existing baselines in multiple evaluation tasks.

## 2  RELATED WORK

While human judgment is widely recognized as the gold standard for evaluating the quality of responses generated by LLMs to open-ended questions, its limitations lie in being both costly and time-consuming. Inspired by the strong instruction-following capabilities of advanced proprietary LLMs, many works (Zhou et al., 2023; Zheng et al., 2023b; Gilardi et al., 2023; Zeng et al., 2023) have explored the use of language models as judges to mimic human evaluators, providing in-depth assessments. Recent research has focused on open-source LLM evaluators, aiming to reduce reliance on proprietary models. PandaLM (Wang et al., 2023b) is trained on LLaMA with instruction tuning, enabling it to conduct pairwise comparisons and provide results along with brief explanations. AUTO-J (Li et al., 2023) trained on 58 real-world scenarios and diverse evaluation protocols, produces well-structured natural language feedback and corresponding scores based on user queries and LLM-generated responses. Prometheus (Kim et al., 2023) emphasizes the importance of reference answers, being trained on data incorporating over 1K different scoring rubrics, thus allowing for adaptation to custom evaluation criteria. Prometheus2 (Kim et al., 2024) further enhances this capability through weight merging, enabling the model to support both direct assessment and pairwise ranking evaluation protocols simultaneously.

However, these models often struggle to deliver precise quantitative scores, typically functioning as task-specific classifiers (Huang et al., 2024). Furthermore, they tend to offer only an overall score, neglecting the need for multi-dimensional quantitative assessments across various question types. Existing studies have highlighted the importance of multi-dimensional evaluation (Li et al., 2024b) and argued that different scenarios require distinct dimensions or metrics to be considered (Li et al., 2024a). In this paper, to the best of our knowledge, we develop the first scenario-aware multi-dimensional evaluator. This model establishes distinct evaluation dimensions customized to specific scenarios and assigns scores to each dimension.

## 3  DATASET CONSTRUCTION

Current pairwise preference datasets for training LLM evaluators, such as HelpSteer (Wang et al., 2023c), UltraFeedback (Cui et al., 2023), and Preference Collection (Kim et al., 2024), have started to incorporate multi-dimensional preference labels and fine-grained evaluation criteria. However, their dimensions remain simplistic and lack scenario-specific customization. To address this, we construct a fine-grained preference dataset spanning diverse scenarios with a comprehensive set of evaluation dimensions, enabling more tailored assessments across various scenarios. Figure 2 illustrates the construction of our dataset, including the scenario and dimension definition, pairwise preference data collection, and fine-grained preference annotation.

### 3.1  SCENARIO AND DIMENSION DEFINITION

**Scenario**  We defined 36 scenarios from the perspective of human needs, categorizing them into five main types based on Maslow's hierarchy: safety, social, cognitive, aesthetic, and self-actualization needs. A detailed description of scenarios is provided in Table A1.

**Dimension**  We designed a total of 42 evaluation dimensions, drawing upon previous studies (Li et al., 2023; 2024a; Sharma et al., 2023). Based on the scenario definitions and descriptions, we in-

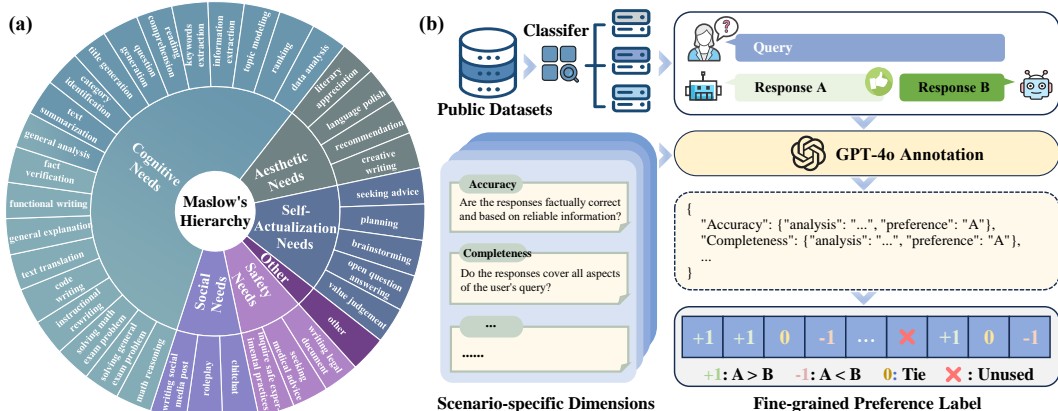

Figure 2: Construction of the fine-grained preference dataset. **(a)** The scenarios in our dataset. **(b)** The pipeline of data collection and annotation.

vited three graduate students with rich experience in natural language processing to pick the required evaluation dimensions for each scenario and assign scenario-specific definitions to each dimension. The comprehensive list of dimensions corresponding to the scenarios is provided in Table A2.

## 3.2 PAIRWISE PREFERENCE DATA COLLECTION

To collect data across a wide range of scenarios, we gathered a large volume of publicly available preference data from multiple sources, including Chatbot Arena Conversations (55K) (Zheng et al., 2023b), Synthetic GPT-J (Havrilla, 2023), Stanford SHP-2(Ethayarajh et al., 2022), HelpSteer-2(Wang et al., 2024b), UltraFeedback (Cui et al., 2023), PKU-SafeRLHF (Ji et al., 2024), and Preference Collection (200K) (Kim et al., 2024). Each dataset contains two model-generated responses from the same instruction (or conversation history), along with overall preference annotations from judges (labeled as win, lose, or tie). To enrich scenario data, we also obtained more data from the Lysms Chat (Zheng et al., 2023a) dataset, which contains 1 million chat instances. To acquire pairwise responses, we used Qwen-2-7B-Inst (Yang et al., 2024) (an LLM not included in the dataset) to generate new responses for each instance, and employed GPT-4o for preference annotation.

Next, we categorize the collected data samples based on their respective scenarios. Given that large-scale scenario annotation by humans or proprietary LLMs is costly, we follow the strategy proposed in (Li et al., 2023) to train a scenario classifier based on Llama3-8b (Dubey et al., 2024). Specifically, we utilize the scenario classification data provided in (Li et al., 2023) as part of our training set, modifying the original scenario labels to align with our definitions. For the missing scenarios, we utilized GPT-4o-mini to annotate 33K instructions from Chatbot Arena Conversations (Zheng et al., 2023b) and select the data that correspond to these scenarios.

Table 1: Statistics of the constructed fine-grained preference dataset

| **Label Distribution** (Label, # of Samples) | | | | | |
|---|---|---|---|---|---|
| Model A Win | 67932 | Model B Win | 58221 | Tie | 9249 |
| **Source Dataset Distribution** (Source, # of Samples) | | | | | |
| Synthetic GPT-J | 3232 | SHP 2 | 11224 | HelpSteer2 | 4033 |
| Lmsys Chat | 21408 | UltraFeedback | 30555 | Chatbot Arena Conversation | 13112 |
| PKU-SafeRLHF | 10095 | Preference Collection | 41743 | | |
| **Scenario Distribution** (Name, # of Samples) | | | | | |
| Title Generation | 2000 | Instructional Rewriting | 2826 | Chitchat | 5500 |
| Language Polish | 2000 | Value Judgement | 4318 | Planning | 5500 |
| Recommendation | 5500 | Roleplay | 5111 | Fact Verification | 3177 |
| General Explanation | 5500 | category identification | 4157 | Creative Writing | 5500 |
| Question Generation | 3852 | Text Summarization | 4033 | Reading Comprehension | 2000 |
| General Analysis | 5500 | Functional Writing | 5500 | Code Writing | 3674 |
| Information Extraction | 3249 | Reasoning | 4453 | Topic Modeling | 2000 |
| Seeking Advice | 5500 | Solving General Exam Problem | 2689 | Open Question | 5500 |
| Solving Math Exam Problem | 2000 | Text-To-Text Translation | 5500 | Ranking | 2000 |
| Data Analysis | 3863 | Writing Social Media Post | 2000 | Brainstorming | 5500 |
| Keywords Extraction | 2000 | Literary Appreciation | 2000 | Seeking Medical Advice | 2000 |
| Safe Experimental Practices | 2000 | Writing Legal Document | 2000 | Other | 5500 |

After labeling scenarios with the classifier, we sampled at least 6K instances for each scenario from the original large dataset. To further ensure label accuracy, we employed GPT-4o-mini to verify the scenario labels and filtered out any data deemed inaccurate. Finally, we balanced the number of samples per scenario between 2K and 5K to maintain similar proportions across all scenarios, with 135,402 data in total.

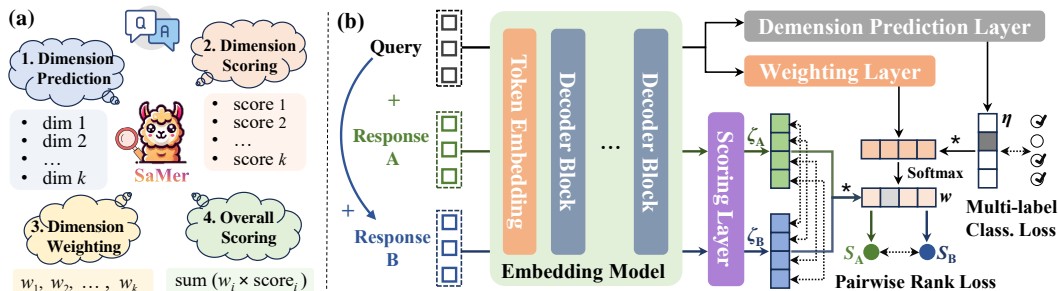

Figure 3: Overview of the proposed SaMer. **(a)** Illustration of the evaluation process of SaMer, which mainly involves four steps: 1) identifying the appropriate evaluation dimensions, 2) scoring the response quality in those dimensions, 3) weighting the contribution of those dimensions, 4) and calculating an overall score through weighted summation. **(b)** Model architecture and training. SaMer consists of a text embedding model and three MLP-based prediction heads (i.e., dimension prediction, scoring, and weighting layers). The training of SaMer involves minimizing the multi-label classification loss as well as multi-dimensional and overall ranking losses.

### 3.3 FINE-GRAINED PREFERENCE ANNOTATION

Based on the scenario labels and scenario-specific dimensions, we utilize GPT-4o to perform fine-grained preference annotation. For each sample, considering that it is more reliable for LLMs to perform pairwise comparisons rather than single-instance evaluations (Liu et al., 2024), we instruct GPT-4o to focus solely on the specified dimension to carefully compare the two model-generated responses and select the better one or declare a tie.

Finally, we constructed a fine-grained preference dataset $\mathcal{D}$ (Table 1). Each sample in $\mathcal{D}$ comprises a query $x$, a pair of responses $(r_A, r_B)$, and an overall preference label $p_o \in \{A, B, \text{Tie}\}$. Additionally, each sample is assigned a scenario label $s$, as well as a set of multi-dimensional fine-grained preference labels $P = \{p_1, p_2, ..., p_{N_s}\}$, where $N_s$ denotes the number of dimensions relevant to the given scenario.

## 4 MODEL

In this section, we present the development of SaMer, utilizing the constructed fine-grained preference dataset. Figure 3 shows the model architecture, training, and evaluation process of SaMer. Specifically, the training process involves dimension prediction via multi-label classification, multi-dimensional scoring via pairwise preference learning, and overall scoring via multi-dimensional weighted summation.

### 4.1 DIMENSION PREDICTION VIA MULTI-LABEL CLASSIFICATION

We first dedicate SaMer to adaptively predict which evaluation dimensions should be considered for each query. To achieve this, we train a dimension predictor using annotated evaluation dimensions in the fine-grained preference dataset $\mathcal{D}$, which selects the appropriate evaluation dimensions based on different scenarios. Specifically, we utilize a frozen pre-trained LLM without its original output projection layer as a text encoder $f_\theta$. Given a query $x$, we feed it into $f_\theta$ and extract the hidden state corresponding to the last token from the final decoder layer, yielding an $h$-dimensional text embedding vector $f_\theta(x) \in \mathbb{R}^h$. We then add a dimension prediction layer (a shallow multi-layer perceptron (MLP)) $\phi_c \in \mathbb{R}^{h \times N}$ on the top of $f_\theta$, which outputs an $N$-dimensional prediction vector $\eta = \phi_c^\top f_\theta(x) \in \mathbb{R}^N$. The training of this MLP can be viewed as a multi-label classification task. To mitigate the potential label imbalance issue, we employ the ZLPR (Zero-bounded Log-sum-exp & Pairwise Rank-based) (Su et al., 2022) loss:

$$\mathcal{L}_{\text{zlpr}} = \log \left(1 + \sum_{i \in P} e^{-\eta_i}\right) + \log \left(1 + \sum_{j \notin P} e^{\eta_j}\right). \tag{1}$$

## 4.2 MULTI-DIMENSIONAL SCORING VIA PAIRWISE PREFERENCE LEARNING

We then leverage the constructed fine-grained preference dataset to achieve multi-dimensional scoring. We introduce a scoring layer $\phi_s \in \mathbb{R}^{h \times N}$ (another MLP) that outputs an $N$-dimensional score vector $\zeta_i = \phi_s^\top f_\theta(x \oplus r_i) \in \mathbb{R}^N$, based on the text embedding extracted from the concatenation of the query $x$ and the response $r_i$, where $i \in \{A, B\}$. Here, we adopt a pairwise preference learning approach. Specifically, for each training sample in $\mathcal{D}$, we first use a mask to exclude dimensions not covered by the preference label set $P$. Then, for each dimension $j$, when the preference labels for responses $r_A$ and $r_B$ are not tied ($p_j \neq 0$) on this dimension, we train them using a margin ranking loss (Cao et al., 2007) with a predefined margin $\gamma$. If a tie occurs ($p_j = 0$), meaning both responses have similar scores on this dimension, we apply a regression loss to make their scores as close as possible. Thus, the multi-dimensional preference loss is formulated as

$$\mathcal{L}_{\text{dim}} = \sum_{j=1}^{N_s} \left[ |p_j| \cdot \max\left(0, -p_j\left(\zeta_{Aj} - \zeta_{Bj}\right) + \gamma\right) + (1 - |p_j|) \cdot \left\|\zeta_{Aj} - \zeta_{Bj}\right\|_2 \right], \quad (2)$$

where the preference label $p_j \in \{+1, -1, 0\}$ represents that response A is superior/inferior/equal to response B in the $j$-th dimension.

## 4.3 OVERALL SCORING VIA MULTI-DIMENSIONAL WEIGHTED SUMMATION

To obtain the overall evaluation score, a straightforward approach is to linearly combine the all of dimensional scores, but this overlooks the fact that the contribution of each dimension varies across different evaluation scenarios. To address this, we introduce a weighting layer $\phi_g \in \mathbb{R}^{h \times N}$, which produces a set of non-negative normalized weights $\{w = \phi_g^\top f_\theta(x) \in \mathcal{R}^N \mid w_i \geq 0, \sum w_i = 1\}$ based on the embedding $f_\theta(x) \in \mathbb{R}^h$ of the query $x$. These weights are then multiplied with the multi-dimensional scores $\zeta$ to obtain the scalar overall score $S$ for the given $x$ and response $r$, i.e., $S = w^\top \zeta$. To train the weighting layer, we employ the previous preference loss again:

$$\mathcal{L}_{\text{o}} = |p_o| \cdot \max\left(0, -p_o\left(S_A - S_B\right) + \gamma\right) + (1 - |p_o|) \cdot \left\|S_A - S_B\right\|_2, \quad (3)$$

where $p_o \in \{+1, -1, 0\}$ denotes the overall preference label between responses A and B.

Finally, we jointly optimize the dimension prediction, scoring and weighting layers of SaMer by minimizing the combination of the aforementioned losses:

$$\min_{\phi_c, \phi_s, \phi_g} \mathbb{E}_{\mathcal{D}} \left(\mathcal{L}_{\text{zlpr}} + \lambda_1 \mathcal{L}_{\text{dim}} + \lambda_2 \mathcal{L}_{\text{o}}\right), \quad (4)$$

where $\lambda_1$ and $\lambda_2$ are the hyper-parameters that balance the contributions of each loss.

## 5 EXPERIMENTS

### 5.1 IMPLEMENTATION OF SAMER

We utilize the Llama-3 8B (Dubey et al., 2024) as the text embedding model and initialize it with the parameters from a Llama-3 8B Reward Model trained by (Wang et al., 2024a). Three MLP layers, specifically $\phi_c$, $\phi_s$, and $\phi_g$, are appended to the embedding model and trained with the loss function specified in Eq. (4), while keeping the embedding frozen. Notably, $\phi_c$ undergoes a warm-up pre-training phase using the loss function $\mathcal{L}_{\text{zlpr}}$ in Eq. (1), aiming to enhance SaMer's scenario-aware multi-dimensional prediction capability.

During training, we set $\gamma = 0.3$ in Eq. (2) and (3), $\lambda_1 = \lambda_2 = 1$ in Eq. (4). Particularly, the impact of $\lambda_1$ and $\lambda_2$ on the performance of SaMer is discussed in the Appendix A4.1. The text embedding dimension $h$ is 4096, consistent with the hidden size of Llama-3-8B, while the dimension $N$ of the three MLP layers is 42, representing the total number of pre-defined dimensions. To efficiently train the model, we leverage the DeepSpeed library(Rasley et al., 2020), Zero Redundancy Optimizer (ZeRO) Stage 2 (Rajbhandari et al., 2020), and FlashAttention2 (Dao, 2023) across 2 NVIDIA GeForce RTX 4090. We adopt AdamW (Loshchilov, 2017) as our optimizer, with $\beta_1 = 0.9$, $\beta_2 = 0.95$, and a weight decay of 0.1. The peak learning rate is set to 5e-5, with 10% warm-up steps, and a cosine decay to 0. We set the batch size to 32 and the maximum sequence length to 8,192. The model is trained for 3 epochs to ensure convergence and optimal performance.

## 5.2 Benchmarks and Metrics

To thoroughly evaluate SaMer, we employ three types of benchmarks: overall-level single rating and pairwise comparison, as well as dimension-level fine-grained comparison.

**Single Rating (Overall)** is a practical evaluation approach as it eliminates the need to prepare a comparison baseline. However, it is inherently challenging since the evaluating language model (LM) must produce scores based on its own internal judgment without external references. We have prepared three single rating benchmarks:

- **Vicuna Bench** (Kim et al., 2023): Adapted from Vicuna (Chiang et al., 2023), this benchmark consists of 80 single-turn chat prompts and 320 total responses generated by four models: WizardLM-13B, Vicuna-13B, Llama2-13B-Chat, and GPT-3.5-Turbo-0613. The responses are scored by GPT-4.
- **FLASK Eval** (Ye et al., 2023): A benchmark contains 200 diverse test prompts and 2K responses from Alpaca-7B, Vicuna-13B, Bard, and GPT-3.5-Turbo-0613. We utilize the human-annotated scores as the reference standard.
- **Feedback Bench** (Kim et al., 2023): The test set of Feedback Collection (Kim et al., 2023) comprising 1K scoring rubrics, 200 instructions, and 1K model-generated responses. To accurately gauge the model's scoring capabilities, we exclude the scoring guidelines and reference answers.

**Pairwise comparison (Overall)** is an effective strategy for evaluating LLMs. By comparing pairs of model-generated responses, it reduces subjectivity in assessment and encourages the identification of finer distinctions. Through pairwise comparisons, we aim to explore the alignment between the evaluator and human preferences. We selected five benchmarks for this purpose:

- **HHH Alignment** (Askell et al., 2021): A dataset of 221 paired responses created by Anthropic, designed to evaluate a model's alignment and preference accuracy across four scenarios: *Helpfulness*, *Harmlessness*, *Honesty*, and *Other*.
- **LLMBar** (Zeng et al., 2023): meta-evaluation benchmark aimed at assessing LLM evaluators' ability to distinguish instruction-following outputs. It includes one Natural subset and four Adversarial subsets: *Neighbor*, *GPTInst*, *GPTOut*, and *Manual*. The Natural subset reflects real-world distributions with objective preferences, while the Adversarial subset contains crafted outputs deviating from instructions.
- **AutoJ Eval**: An in-domain test set from (Li et al., 2023), consisting of 58 prompts and 1,392 response pairs, with preferences labeled by human annotators as win, tie, or lose.
- **AlpacaEval** (Dubois et al., 2024): A benchmark reconstructed from the Alpac-Farm (Dubois et al., 2023) dataset, containing 805 test prompts from five data sources. We use a version scored by GPT-4, which includes 13 model responses and 10,463 pairs.
- **Preference Bench** (Kim et al., 2024): An in-domain test set of the Prometheus model, containing 200 prompts and 2,000 response pairs. This dataset is generated by combining responses from five different models in the Feedback Collection (Kim et al., 2023).

**Fine-Grained Comparison (Multi-dimensional)** is a complex evaluation task that requires an evaluator to compare pairs of responses based on a specified dimension and produce preference judgments that are exclusive to that dimension. The model's effectiveness on this task hinges on its ability to deeply understand the given dimension while remaining entirely uninfluenced by others. To assess this capability, we use a concealed test set with 10 samples per scenario derived from the multi-scenario, multi-dimensional fine-grained preference data gathered in Section 3.3, which we term as **MD-Eval** that encompasses a total of 360 human-verified samples.

## 5.3 Baselines

We employed *proprietary LLM baselines*, including OpenAI's GPT-4o-2024-05-13 (OpenAI, 2024a), GPT-4o-mini-2024-07-18 (OpenAI, 2024b), and Anthropic's Claude-3.5-Sonnet-2024-06-

Table 2: Evaluation results on three single rating benchmarks. **Bold results** and underline results indicate the best and the second-best results among the open-source models, respectively. Same as below.

| Evaluator | Vicuna Bench | | | FLASK Eval | | | Feedback Bench | | |
|---|---|---|---|---|---|---|---|---|---|
| | Pearson | Spearman | Kendall-Tau | Pearson | Spearman | Kendall-Tau | Pearson | Spearman | Kendall-Tau |
| GPT-4o-mini | 0.456 | 0.323 | 0.286 | 0.493 | 0.449 | 0.371 | 0.752 | 0.764 | 0.671 |
| Claude-3.5-Sonnet | 0.489 | 0.326 | 0.286 | 0.420 | 0.400 | 0.322 | 0.726 | 0.726 | 0.630 |
| GPT-4o | 0.434 | 0.306 | 0.263 | 0.501 | 0.478 | 0.389 | 0.735 | 0.731 | 0.636 |
| Llama-2-7B-Chat | 0.021 | 0.042 | 0.037 | 0.092 | 0.072 | 0.059 | 0.492 | 0.553 | 0.488 |
| Llama-2-13B-Chat | 0.072 | 0.014 | 0.013 | 0.217 | 0.133 | 0.113 | 0.569 | 0.538 | 0.468 |
| Llama-3-8B-Inst | -0.007 | -0.008 | -0.007 | 0.091 | 0.072 | 0.062 | 0.464 | 0.489 | 0.434 |
| Llama-3.1-8B-Inst | 0.298 | 0.188 | 0.167 | 0.336 | 0.269 | 0.228 | 0.623 | 0.660 | 0.586 |
| Mistral-7B-Inst | 0.190 | 0.117 | 0.104 | 0.243 | 0.179 | 0.154 | 0.525 | 0.531 | 0.469 |
| AutoJ-13B | 0.360 | 0.364 | 0.317 | 0.458 | 0.384 | 0.312 | 0.609 | 0.605 | 0.523 |
| Prometheus-7B | 0.413 | 0.416 | **0.354** | 0.246 | 0.206 | 0.161 | - | - | - |
| Prometheus-13B | 0.268 | 0.272 | 0.237 | 0.354 | 0.314 | 0.255 | - | - | - |
| Prometheus2-7B | 0.267 | 0.254 | 0.219 | 0.335 | 0.259 | 0.209 | - | - | - |
| ArmoRM-8B | 0.446 | 0.396 | 0.300 | 0.359 | 0.315 | 0.225 | 0.749 | 0.754 | 0.607 |
| **SaMer-8B** | **0.476** | **0.458** | **0.354** | **0.515** | **0.468** | **0.345** | **0.790** | **0.783** | **0.631** |

20 (Anthropic, 2024), alongside *open-source LLM baselines* such as Llama-2-7B/13B-Chat (Touvron et al., 2023), Llama-3-8B-Instruct (Dubey et al., 2024), Llama-3.1-8B-Instruct (Dubey et al., 2024), and Mistral-7B-Instruct-v0.2 (Jiang et al., 2023). Given that these models were not specifically trained for evaluation tasks but exhibit strong instruction-following capabilities, we used prompts from LLMBar (Zeng et al., 2023) to assess their performance on evaluation tasks. For *state-of-the-art LLM evaluator baselines*, we adopted state-of-the-art models including AutoJ-13B (Li et al., 2023), Prometheus-7B/13B (Kim et al., 2023), Prometheus2-7B (Kim et al., 2024), and ArmoRM-8B (Wang et al., 2024a). To ensure a fair comparison, we used each model's original prompt templates and excluded the reference answer module from the templates of Prometheus series.

## 5.4 EVALUATION RESULTS

To demonstrate our SaMer's adaptability across various evaluation modes, we conducted three sets of experiments: single ratings, pairwise comparisons, as well as multi-dimensional fine-grained comparisons. As our experiments focus on investigating the generalizability of evaluators across diverse evaluation scenarios, we have not included the held-in results, such as Prometheus' performance in Feedback Bench.

In the **single rating** (Table 2) tasks, SaMer demonstrated significant improvements across three single rating benchmarks, particularly in FLASK Eval. It is noteworthy that SaMer exhibits comparable performance and even surpasses proprietary models, including GPT-4o/4o-mini and Claude-3.5-Sonnet (detailed discussion is illustrated in the Appendix A4.3). However, one can observe that most models (including SaMer) did not achieve a correlation coefficient exceeding 0.5 with the annotated labels on Vicuna Bench and FLASK, indicating the challenging nature of these benchmarks and the complexity of aligning model evaluations with human ratings.

In the **pairwise comparison** tasks (Tables 3 and 4), SaMer achieved top performance in 9 out of 15 tasks and ranked second in the remaining 6 tasks among the open-source models. This strong performance can be partly attributed to the robust ArmoRM backbone, which also delivered leading results across multiple tasks. However, we highlight the effectiveness of our strategy, as evidenced by SaMer's notable improvements on AlpacaEval. Despite that the proprietary models generally outperform all open-source models in these evaluations, SaMer showcases competitive results by outperforming many open-source baselines and coming close to the proprietary models' performance in several tasks, highlighting its robust adaptability.

In the **fine-grained comparison** tasks (Table 5), the results show that dimension-level accuracy was generally lower than overall accuracy for most evaluators, underscoring the challenge of accurately evaluating responses on specific dimensions. An intriguing observation was the performance drop in Llama-2-13B-Chat compared to its 7B counterpart, suggesting that increasing model parameters

Table 3: Evaluation results on four pairwise comparison benchmarks

| Evaluator | HHH Alignment | | | | | AutoJ Eval | | Preference Bench | AlpacaEval | |
| --- | --- | --- | --- | --- | --- | --- | --- | --- | --- | --- |
| | Help. | Harm. | Hon. | Other | Total Avg. | w/o TIE | w/ TIE | OOD | w/o TIE | w TIE |
| GPT-4o-mini | 89.83 | 91.38 | 75.41 | 93.02 | 86.88 | 79.20 | 59.34 | 88.85 | 82.01 | 83.54 |
| Claude-3.5-Sonnet | 91.53 | 94.83 | 88.52 | 93.02 | 91.86 | 76.64 | 61.68 | 85.10 | 73.70 | 76.02 |
| GPT-4o | 89.83 | 94.83 | 88.52 | 93.02 | 90.95 | 76.05 | 58.33 | 84.10 | 76.58 | 78.61 |
| Llama-2-7B-Chat | 59.32 | 71.43 | 45.90 | 53.66 | 56.12 | 48.99 | 35.99 | 57.04 | 50.56 | 46.15 |
| Llama-2-13B-Chat | 71.43 | 76.47 | 61.22 | 71.43 | 68.79 | 54.95 | 40.62 | 63.91 | 54.39 | 50.62 |
| Llama-3-8B-Inst | 79.66 | 80.70 | 73.77 | 88.37 | 80.00 | 62.51 | 48.49 | 75.55 | 66.90 | 69.45 |
| Llama-3.1-8B-Inst | 83.05 | 84.62 | 78.69 | 88.10 | 83.18 | 71.61 | 52.70 | 79.80 | 74.24 | 74.38 |
| Mistral-7B-Inst | 71.19 | 81.03 | 67.21 | 74.42 | 73.30 | 58.74 | 43.34 | 63.40 | 56.08 | 56.40 |
| AutoJ-13B | 70.49 | 82.76 | 77.97 | 72.09 | 76.02 | - | - | 77.43 | 77.72 | 72.59 |
| Prometheus-7B | 50.85 | 43.10 | 54.10 | 48.84 | 49.32 | 47.20 | 47.63 | 79.90 | 43.72 | 48.57 |
| Prometheus-13B | 66.10 | 48.28 | 34.43 | 65.12 | 52.49 | 46.71 | 48.64 | 76.80 | 40.03 | 45.23 |
| Prometheus2-7B | 83.05 | 75.86 | 63.93 | 76.74 | 74.66 | 74.78 | 55.72 | - | 74.05 | 75.00 |
| ArmoRM-8B | **89.83** | **93.10** | **78.69** | 93.02 | **88.24** | 76.74 | 57.18 | **86.90** | 76.74 | 78.76 |
| **SaMer-8B** | 88.14 | 89.66 | **78.69** | **95.35** | 87.33 | **77.72** | **58.69** | 83.50 | **81.38** | **83.00** |

Table 4: Evaluation results on the LLMBar benchmark

| Evaluator | GPTInst | GPTOut | Manual | Neighbor | Natural |
| --- | --- | --- | --- | --- | --- |
| GPT-4o-mini | 83.70 | 65.96 | 63.04 | 67.16 | 91.00 |
| Claude-3.5-Sonnet | 88.04 | 61.70 | 78.26 | 85.07 | 92.00 |
| GPT-4o | 88.04 | 76.60 | 78.26 | 77.61 | 99.00 |
| Llama-2-7B-Chat | 48.35 | 46.81 | 41.30 | 43.61 | 58.00 |
| Llama-2-13B-Chat | 33.77 | 47.83 | 31.82 | 29.13 | 70.10 |
| Llama-3-8B-Inst | 39.13 | 55.32 | 41.30 | 21.64 | 78.00 |
| Llama-3.1-8B-Inst | 43.48 | 55.32 | 43.48 | 33.08 | 83.00 |
| Mistral-7B-Inst | 51.09 | 46.81 | 45.65 | 45.52 | 76.00 |
| AutoJ-13B | 23.91 | 50.00 | 26.67 | 23.48 | 71.13 |
| Prometheus-7B | 15.22 | 36.17 | 34.78 | 17.16 | 48.00 |
| Prometheus-13B | 14.13 | 46.81 | 28.26 | 15.67 | 59.00 |
| Prometheus2-7B | 29.35 | 58.70 | 37.78 | 22.39 | 77.00 |
| ArmoRM-8B | **77.17** | 63.83 | **69.57** | 67.16 | **93.00** |
| **SaMer-8B** | 54.35 | 65.96 | 69.57 | 86.57 | 84.00 |

Table 5: Evaluation results on the MD-Eval dataset

| Evaluator | Dim Acc. | Overall Acc. |
| --- | --- | --- |
| GPT-4o-mini | 72.99 | 78.00 |
| Claude-3.5-Sonnet | 61.63 | 74.15 |
| GPT-4o | - | - |
| Llama-2-7B-Chat | 53.13 | 53.58 |
| Llama-2-13B-Chat | 48.47 | 53.47 |
| Llama-3-8B-Inst | 64.96 | 66.67 |
| Llama-3.1-8B-Inst | 73.13 | 71.91 |
| Mistral-7B-Inst | 55.70 | 62.80 |
| AutoJ-13B | 53.58 | 61.12 |
| Prometheus-7B | 60.22 | 38.33 |
| Prometheus-13B | 64.96 | 43.67 |
| Prometheus2-7B | 67.11 | 71.24 |
| ArmoRM-8B | - | 79.33 |
| **SaMer-8B** | **75.67** | **82.33** |

does not necessarily lead to better fine-grained evaluation capabilities. In contrast, SaMer achieved notable improvements, with a +10.7 increase in dimension-level accuracy and a +15.7 increase in overall accuracy compared to the Llama-3-8B-Inst, the original backbone of SaMer. Moreover, when compared to proprietary models, SaMer outperformed both GPT-4o-mini and Claude-3.5-Sonnet. This result underscores the strength and effectiveness of our training methodology in enhancing multi-dimensional, scenario-aware evaluations.

## 5.5 INTERPRETABLE EVALUATIONS

In addition to its robust performance across various benchmarks, SaMer excels in providing interpretable evaluations by offering detailed dimension-specific scores and weights. These fine-grained assessments enable a deeper understanding of how each dimension contributes to the overall quality, allowing us to identify the dominant factors in this evaluation.

SaMer exhibits scenario-aware adaptability, effectively identifying contextually appropriate evaluation dimensions for queries that may not have explicit scenario labels. This capability is exemplified in Fig. 4, which shows the weights assigned by SaMer across three distinct scenarios: *Creative Writing*, *Math Reasoning*, and *Writing Legal Document*. These weights indicate the relative importance of each dimension within the evaluation process. For Creative Writing, SaMer assigns the greatest importance to the creativity dimension, followed sequentially by logic, relevance, harmlessness, and style, which aligns closely with the essential attributes of creative writing. In contrast, for Math Reasoning, the emphasis shifts towards the reasoning process and outcome, with logic, accuracy, clarity, and step-by-step explanation emerging as the predominant dimensions. When evaluating the

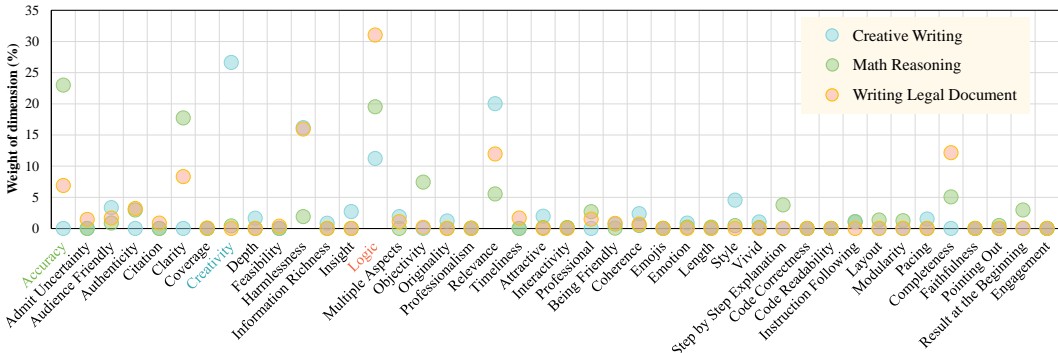

Figure 4: Average dimension weights of SaMer when evaluating scenarios in the MD-Eval dataset, including *Creative Writing*, *Math Reasoning*, and *Writing Legal Document*. The most preferred attribute for each category is highlighted in the corresponding color: Creativity for Creative Writing, Accuracy for Math Reasoning, and Logic for Writing Legal Document.

Writing Legal Document scenario, logic is identified as the most critical dimension, followed by considerations of harmlessness and clarity, reflecting the distinct requirements of legal writing.

To demonstrate SaMer's accuracy in dimension prediction, we compared its performance with generative LLM baselines on the **dimension selection** task. The full set of 42 dimensions was fixed to ensure consistency across models, with baselines guided by prompts to select appropriate dimensions for each scenario. The evaluation was on both in-domain (ID, MD-Eval) and out-of-domain (OOD, Auto-J Eval) (Li et al., 2023) datasets. For OOD, 108 unique examples were selected after filtering out overlapping scenarios. Precision and Recall were used for ID evaluation, while OOD evaluation was based on the win rate against SaMer (the proportion of cases where a baseline outperformed SaMer according to human judgment).

Table 6: Performance comparison of SaMer and LLM baselines on the *dimension selection* task (ID and OOD).

| Evaluator | ID (%) | | OOD (%) |
|---|---|---|---|
| | Precision | Recall | Win Rate |
| GPT-4o | 63.42 | 38.10 | 48.15 |
| GPT-4o-mini | 57.61 | 37.89 | 37.65 |
| Mistral-7B-Inst | 42.73 | 23.82 | 14.42 |
| Llama2-7B-Chat | 27.78 | 37.36 | 10.84 |
| Llama3-8B-Inst | 39.81 | 36.37 | 21.61 |
| Llama3.1-8B-Inst | 34.96 | 60.55 | 28.06 |
| **SaMer-8B** | **74.84** | **72.33** | - |

The results in Table 6 show that SaMer outperforms the baselines. In the ID evaluation, SaMer achieves 74.84% Precision and 72.33% Recall, while most baselines fail to exceed 50% Precision and 40% Recall, indicating current LLMs' limitations. Among the baselines, GPT-4o performs relatively well with high precision but low recall, suggesting a more selective (and less comprehensive) subset of dimensions. In the OOD evaluation, SaMer surpasses GPT-4o, whose win rate is below 50%, showing that SaMer aligns better with human preferences.

## 6 CONCLUSION

In this work, we propose SaMer, a scenario-aware multi-dimensional evaluator designed to provide fine-grained and interpretable assessments of LLM-generated responses. By dynamically identifying and prioritizing relevant evaluation dimensions for different query scenarios, SaMer enables more nuanced and context-sensitive evaluations compared to conventional fixed-dimension approaches. Extensive experiments across single rating and pairwise comparison benchmarks validate the model's adaptability, showing that SaMer outperforms existing baselines while offering transparent and interpretable assessments through detailed dimension-level scores and weights.

Despite its strengths, SaMer has some limitations. First, its performance depends on the quality and diversity of the fine-grained preference dataset, which, while comprehensive, may not cover all possible real-world scenarios. Additionally, the model's capacity to interpret complex and overlapping dimensions requires further enhancement. For future work, expanding the dataset to include broader scenarios and refining SaMer to better handle contextually ambiguous queries would improve its robustness. Furthermore, SaMer can serve as a reward model for reinforcement learning, guiding LLMs to optimize not just overall quality but also specific dimensions that match human preferences, enhancing response quality in a targeted way.

ACKNOWLEDGEMENTS

This work is supported by Zhejiang Provincial "Jianbing" "Lingyan" Research and Development Program of China (2024C01135, 2025C01097), National Natural Science Foundation of China (62301480, 62302433, U23A20496), Hangzhou West Lake Pearl Project Leading Innovative Youth Team Project (TD2023017) and Alibaba Group through Alibaba Innovative Research Program.

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

## A1 SCENARIO AND DIMENSION DEFINITION

In this section, we provide a detailed presentation of the scenarios and evaluation dimensions. Table A1 lists the 36 specific scenarios along with their definitions (or descriptions), while Table A2 presents the human-verified evaluation dimensions required for each scenario. Since the definitions of the dimensions are scenario-specific, we have provided a dedicated set of evaluation dimension definitions for each scenario in Section A1.1.

### A1.1 SCENARIO-SPECIFIC EVALUATION DIMENSION DEFINITIONS

---

**Writing Legal Document**

1. `Accuracy`: The dialogue must provide precise and correct information, ensuring that all legal terms, conditions, and clauses are correctly stated and legally sound.
2. `Admit Uncertainty`: The dialogue should acknowledge any areas of uncertainty or lack of information, providing guidance on where to seek further clarification or legal advice.
3. `Authenticity`: The dialogue should reflect genuine legal standards and practices, using appropriate legal terminology and referencing actual legal principles and precedents.
4. `Citation`: The dialogue should include references to relevant laws, statutes, regulations, or legal cases to support the information provided, ensuring the document"s credibility and legal validity.
5. `Clarity`: The dialogue must be clear and understandable, avoiding overly complex language or jargon that could confuse the reader. Legal terms should be explained where necessary.
6. `Completeness`: The dialogue should cover all necessary aspects of the legal document, ensuring that no critical information or sections are omitted.
7. `Harmlessness`: The dialogue must ensure that the content is not harmful or misleading, providing accurate legal information that does not put the user at risk of legal issues.
8. `Layout`: The dialogue should guide the user in structuring the legal document properly, ensuring that it follows a logical and accepted format for legal texts.
9. `Logic`: The dialogue should follow a logical sequence, ensuring that the legal arguments and clauses are presented in a coherent and rational manner.
10. `Multiple Aspects`: The dialogue should consider various aspects of the legal issue at hand, providing a comprehensive view that addresses different angles and potential implications.
11. `Objectivity`: The dialogue should maintain an objective tone, presenting information and advice based on legal standards and facts rather than personal opinions or biases.
12. `Professional`: The dialogue should exhibit a high level of professionalism, reflecting the formal and serious nature of legal document drafting, and ensuring that all advice is sound and reliable.

---

**Seeking Medical Advice**

1. `Accuracy`: The information provided should be medically accurate, based on current medical knowledge and guidelines.
2. `Admit Uncertainty`: The dialogue should acknowledge when the information is uncertain or when further professional consultation is necessary.
3. `Authenticity`: The advice should be genuine and trustworthy, reflecting real medical knowledge and practices.
4. `Citation`: Whenever possible, the information should be backed by credible sources or references to medical literature.
5. `Clarity`: The language used should be clear and easy to understand, avoiding medical jargon unless it is explained.
6. `Completeness`: The response should cover all relevant aspects of the medical query, providing a thorough answer.
7. `Coverage`: The advice should address the full scope of the user"s question, considering various facets of the medical issue.
8. `Feasibility`: The recommendations given should be practical and achievable for the user.
9. `Harmlessness`: The information should be safe, avoiding any advice that could potentially harm the user.
10. `Logic`: The advice should follow a logical sequence, with well-reasoned explanations and conclusions.
11. `Multiple Aspects`: The dialogue should consider different perspectives and aspects of the medical issue, such as symptoms, potential causes, and treatment options.
12. `Objectivity`: The advice should be unbiased and based on factual information rather than personal opinions.
13. `Professionalism`: The tone and content should reflect a professional attitude, maintaining a respectful and serious approach to the medical issue.
14. `Relevance`: The response should be directly relevant to the user"s specific medical query or concern.
15. `Style`: The communication style should be appropriate for a medical context, maintaining a balance between professionalism and empathy.
16. `Timeliness`: The advice should be current, reflecting the latest medical guidelines and research.

---

## Inquire Safe Experimental Practices

1. `Accuracy`: The information provided must be factually correct and based on established scientific principles and safety standards.
2. `Authenticity`: The advice and guidelines should be genuine and reflect real-world practices and regulations.
3. `Citation`: Relevant sources, such as scientific literature, regulatory guidelines, or expert opinions, should be cited to support the information provided.
4. `Clarity`: The advice should be communicated in a clear and understandable manner, avoiding technical jargon unless it is explained.
5. `Completeness`: The response should cover all necessary aspects of safe experimental practices, including preparation, execution, and post-experiment procedures.
6. `Coverage`: The response should address a wide range of safety considerations, including but not limited to chemical, biological, physical, and environmental hazards.
7. `Feasibility`: The recommended practices should be practical and achievable within the constraints of typical experimental settings.
8. `Harmlessness`: The information should ensure that no harmful or dangerous advice is given, prioritizing the well-being of individuals and the environment.
9. `Logic`: The advice should follow a logical sequence, with clear reasoning and rationale behind each recommendation.
10. `Objectivity`: The information should be unbiased and based on objective evidence rather than personal opinions or anecdotal experiences.
11. `Professionalism`: The tone and content should reflect a high level of expertise and professionalism, suitable for a scientific or technical audience.
12. `Relevance`: The advice should be directly relevant to the specific type of experiment or safety concern being inquired about.

## Chitchat

1. `Attractive`: The response is engaging and able to capture the user's interest, making the conversation enjoyable and lively.
2. `Audience Friendly`: The response is considerate of the user's needs and feelings, fostering a positive and inclusive interaction.
3. `Being Friendly`: The character displays a friendly attitude and tone, making the user feel welcomed and comfortable.
4. `Coherence`: The response flows naturally and logically, with responses that smoothly connect to previous and following statements.
5. `Creativity`: The character demonstrates unique and imaginative thinking, contributing original and interesting topics to the conversation.
6. `Emojis`: Emojis are appropriately used to enhance the expression, adding emotional color and fun without being excessive or inappropriate.
7. `Emotion`: The character expresses genuine and strong emotions, making the conversation more relatable and engaging.
8. `Harmlessness`: The response ensures that no content is harmful or offensive, maintaining a safe and respectful environment.
9. `Interactivity`: The response promotes active participation and engagement, encouraging the user to contribute to the conversation.
10. `Length`: The responses are of appropriate length, providing sufficient detail without being overly long or too brief.
11. `Logic`: The response follows a logical sequence, with coherent explanations and sound reasoning.
12. `Relevance`: The responses are directly relevant to the user's statements and the overall topic of the conversation.
13. `Style`: The character maintains a consistent and appropriate style, whether formal or informal, as suited to the user's chitchat context.
14. `Vivid`: The response is vivid and lively, using concrete and imaginative language to make the conversation more visually and emotionally engaging.

## Roleplay

1. `Attractive`: The character is engaging and able to capture the user's attention and interest. The dialogue is novel and interesting, sparking the user's curiosity and desire to participate.

2. `Audience Friendly`: The character considers the user's needs and feelings, providing a positive interaction experience. It shows respect and understanding towards the user, with information and interaction methods suitable for the user's level and interests.

3. `Being Friendly`: The character displays a friendly attitude and tone. The language and behavior are gentle and polite, making the user feel welcome and comfortable.

4. `Coherence`: The content of the dialogue is logically clear and connected. Responses naturally link with the previous and following context, avoiding jumps or breaks.

5. `Creativity`: The character demonstrates unique ideas and innovative expressions. The dialogue includes novel and interesting viewpoints or expressions, bringing surprise and inspiration to the user.

6. `Emojis`: Emojis are appropriately used in the dialogue to enhance the expression. They are used fittingly, adding emotional color and fun without being excessive or inappropriate.

7. `Emotion`: The character can express genuine and strong emotions. Emotional expressions are sincere and natural, resonating with the user.

8. `Harmlessness`: The character's words and actions do not cause harm or offense to the user or others. It avoids offensive, discriminatory, or harmful statements and behaviors, ensuring the dialogue is safe and respectful.

9. `Instruction Following`: The character accurately understands and executes the user's instructions. It accurately comprehends the user's requirements without defying or rejecting them.

10. `Interactivity`: The character engages in active and effective interaction with the user. It maintains the interactivity of the dialogue through questions, feedback, and other means, promoting user participation.

11. `Logic`: The character's words and actions are logically sound, clearly expressing viewpoints. Responses are well-organized, with reasonable reasoning processes, avoiding contradictions.

12. `Relevance`: The character's responses are closely related to the dialogue topic and the user's needs. Answers directly address the user's questions or topics, avoiding digressions or irrelevant content.

13. `Style`: The character maintains a consistent response style that matches the expected tone and manner. It keeps a consistent tone and style that fits the user's requested character language style.

14. `Vivid`: The character's expressions are vivid and lively, enhancing the attractiveness of the dialogue. It uses concrete and vivid descriptions and language, making the dialogue more visually and emotionally engaging.

## Writing Social Media Post

1. `Attractive`: The post captures the audience''s attention with compelling visuals, interesting content, or engaging language, making it stand out in a crowded feed.

2. `Audience Friendly`: The post is considerate of the audience''s preferences, interests, and needs, ensuring it resonates well with them.

3. `Being Friendly`: The tone of the post is approachable and warm, making the audience feel welcomed and valued.

4. `Clarity`: The message is clear and easy to understand, avoiding ambiguity and confusion.

5. `Coherence`: The post flows logically and maintains a consistent theme or message throughout.

6. `Creativity`: The post shows originality and imaginative thinking, offering unique content that differentiates it from others.

7. `Emojis`: Emojis are used appropriately to enhance the message, add emotional nuance, and make the post more visually engaging without overwhelming the text.

8. `Emotion`: The post conveys genuine and strong emotions, making it more relatable and engaging for the audience.

9. `Harmlessness`: The content is safe and respectful, avoiding any offensive or harmful language or imagery.

10. `Length`: The post is concise and to the point, providing sufficient information without being overly lengthy or too brief.

11. `Logic`: The post follows a logical structure, with ideas presented in a coherent and rational order.

12. `Originality`: The content is unique and not a mere repetition of existing posts, offering fresh perspectives or information.

13. `Style`: The post maintains a consistent and appropriate style, whether formal or informal, tailored to the platform and audience.

14. `Timeliness`: The post is relevant to current events, trends, or the audience''s immediate interests, making it timely and topical.

15. `Vivid`: The post uses vivid and descriptive language or visuals to create a strong impression and engage the audience''s senses.

Table A1: Detailed description for each scenario in our dataset.

| Safety Needs | |
| --- | --- |
| writing legal document | Writing legal documents is a formal and precise process that involves drafting, reviewing, and finalizing legal texts such as contracts, wills, deeds, and legal agreements. |
| seeking medical advice | Seeking medical advice involves a conversation where an individual seeks information or guidance regarding health-related concerns, symptoms, treatments, or medical conditions. |
| inquire safe experimental practices | Inquiring about safe experimental practices involves seeking information, guidelines, and advice on how to conduct scientific or technical experiments in a manner that ensures the safety of all participants, minimizes risks, and adheres to established standards and regulations. |

| Social Needs | |
| --- | --- |
| chitchat | Chitchat is casual, informal conversation focused on social interaction rather than exchanging detailed information or solving problems. |
| roleplay | Role play dialogue is an interactive simulation where participants assume specific roles to mimic real-life scenarios for training, education, or entertainment purposes. |
| writing social media post | Writing a social media post involves crafting a message intended for sharing on social platforms like Facebook, Twitter, Instagram, etc. |

| Cognitive Needs | |
| --- | --- |
| reasoning | reasoning involves logical and systematic thinking to solve problems. |
| solving general exam problem | Solving a general exam problem involves providing an answer through reasoning, logic, conceptual understanding, or qualitative analysis. |
| solving math exam problem | It requires understanding mathematical concepts, applying appropriate methods, and providing clear, step-by-step explanations to arrive at accurate solutions. |
| instructional rewriting | Instructional rewriting involves modifying or enhancing a given text or response to improve its clarity, accuracy, coherence, or other attributes while faithfully adhering to the original instructions or intent. |
| code writing | Code writing involves creating, editing, and debugging code to solve problems, implement features, or automate tasks. |
| text translation | Text-to-text translation is the process of converting text from one language to another while preserving the meaning, tone, and context of the original text. |
| general explanation | general explanation is the act of providing clear, detailed, and understandable explanations on a wide range of topics, aiming to inform, clarify, and educate the audience effectively. |
| functional writing | Functional writing in a response context refers to the exchange of information or instructions in a clear, precise, and efficient manner to achieve a specific purpose. |
| fact verification | Verifying fact in a dialogue context involves confirming the accuracy and reliability of information presented during the conversation. |
| general analysis | Analyzing general is a type of dialogue focused on examining, interpreting, and understanding various subjects or data in a broad and non-specific manner. |
| text summarization | Text summarization is the process of condensing a piece of text to a shorter version, retaining the most important information and meaning. |
| category identification | Category identification involves accurately classifying or identifying one or multiple objects provided by the user into predefined specific categories. |
| title generation | Title generation involves creating an appropriate and compelling title for a given text or based on a description of a work. |
| question generation | Question generation is the process of creating questions based on given content or context. |
| reading comprehension | Reading comprehension involves answering questions that can be directly answered by information contained within the attached passage. |
| keywords extraction | Keywords extraction involves identifying and extracting the most important and relevant keywords from a given piece of text. |
| information extraction | Information extraction involves extracting specific categories of information as specified by the user from a given piece of text. |
| topic modeling | Topic modeling involves extracting high-level topics or themes from a given text to identify and summarize the main subjects discussed. |
| ranking | Ranking involves sorting a set of items based on specified criteria. |
| data analysis | Data analysis refers to the process of inspecting, cleansing, transforming, and modeling data with the goal of discovering useful information, drawing conclusions, and supporting decision-making. |

| Aesthetic Needs | |
| --- | --- |
| literary appreciation | Literary appreciation is the analysis and evaluation of literary works, focusing on their artistic qualities, themes, and stylistic elements. |
| language polish | Language polish refers to the process of refining and enhancing a piece of text to improve its clarity, readability, and overall quality. |
| recommendation | A recommendation dialogue is a conversation where one party suggests products, services, or actions to another based on their preferences, needs, or interests. |
| creative writing | Creative writing is the art of crafting stories, poetry, and other literary works that emphasize imaginative and original expression. |

| Self-Actualization Needs | |
| --- | --- |
| seeking advice | Seeking advice is a dialogue scenario where one participant requests guidance, recommendations, or solutions to a specific problem or situation. |
| planning | Planning in dialogue refers to conversations where the primary goal is to organize, strategize, and prepare for future actions or events. |
| brainstorming | Brainstorming is a collaborative and creative process where participants generate a wide range of ideas and solutions to a particular problem or challenge. |
| open question answering | Open question dialogue involves queries that cannot be answered with a simple "yes" or "no". |
| value judgement | Value Judgement is the process of evaluating or making decisions based on personal beliefs, values, or standards. |

| Other | |
| --- | --- |
| Other | This is an unlabeled scenario, involves engaging in open-ended conversations, providing information, answering questions, and assisting with various requests based on user instructions. |

Table A2: The evaluation dimensions for each scenario in our dataset

| | |
|---|---|
| **Safety Needs** | |
| writing legal document | Accuracy, Admit Uncertainty, Authenticity, Citation, Clarity, Harmlessness, Logic, Multiple Aspects, Objectivity, Professional, Layout, Completeness |
| seeking medical advice | Accuracy, Admit Uncertainty, Authenticity, Citation, Clarity, Completeness, Coverage, Feasibility, Harmlessness, Logic, Multiple Aspects, Objectivity, Professionalism, Relevance, Style, Timeliness |
| inquire safe exper-imental practices | Accuracy, Authenticity, Citation, Clarity, Completeness, Coverage, Feasibility, Harmlessness, Logic, Objectivity, Professionalism, Relevance |
| **Social Needs** | |
| chitchat | Audience Friendly, Creativity, Harmlessness, Logic, Relevance, Attractive, Interactivity, Being Friendly, Coherence, Emojis, Emotion, Length, Style, Vivid |
| roleplay | Audience Friendly, Creativity, Harmlessness, Logic, Relevance, Attractive, Interactivity, Being Friendly, Coherence, Emojis, Emotion, Style, Vivid, Instruction Following |
| writing social media post | Audience Friendly, Clarity, Creativity, Harmlessness, Logic, Originality, Timeliness, Attractive, Being Friendly, Coherence, Emojis, Emotion, Length, Style, Vivid |
| **Cognitive Needs** | |
| reasoning | Accuracy, Authenticity, Clarity, Logic, Objectivity, Relevance, Professional, Step by Step Explanation, Instruction Following, Layout, Modularity, Completeness, Pointing Out, Result at the Beginning |
| solving general exam problem | Accuracy, Audience Friendly, Authenticity, Clarity, Feasibility, Harmlessness, Logic, Objectivity, Relevance, Timeliness, Professional, Step by Step Explanation, Completeness |
| solving math exam problem | Accuracy, Authenticity, Clarity, Logic, Objectivity, Relevance, Professional, Step by Step Explanation, Instruction Following, Layout, Modularity, Completeness, Pointing Out, Result at the Beginning |
| instructional rewriting | Accuracy, Authenticity, Clarity, Creativity, Harmlessness, Logic, Relevance, Coherence, Length, Style, Instruction Following, Completeness, Faithfulness, Pointing Out |
| code writing | Accuracy, Clarity, Feasibility, Harmlessness, Logic, Professional, Style, Step by Step Explanation, Code Correctness, Code Readability, Instruction Following, Layout, Modularity |
| text translation | Accuracy, Authenticity, Clarity, Harmlessness, Logic, Objectivity, Professionalism, Relevance, Coherence, Style, Instruction Following, Completeness, Faithfulness |
| general explanation | Accuracy, Admit Uncertainty, Audience Friendly, Authenticity, Citation, Clarity, Coverage, Depth, Harmlessness, Logic, Multiple Aspects, Objectivity, Professionalism, Relevance, Coherence, Step by Step Explanation, Instruction Following |
| functional writing | Accuracy, Authenticity, Citation, Clarity, Coverage, Creativity, Depth, Harmlessness, Insight, Logic, Professionalism, Relevance, Coherence, Length, Style, Instruction Following, Layout, Completeness, Faithfulness |
| fact verification | Accuracy, Admit Uncertainty, Authenticity, Citation, Clarity, Harmlessness, Information Richness, Objectivity, Relevance, Timeliness, Instruction Following, Completeness, Faithfulness, Result at the Beginning |
| general analysis | Accuracy, Admit Uncertainty, Audience Friendly, Authenticity, Citation, Clarity, Coverage, Creativity, Depth, Feasibility, Harmlessness, Information Richness, Insight, Logic, Multiple Aspects, Objectivity, Originality, Professionalism, Relevance, Timeliness |
| text summarization | Accuracy, Citation, Clarity, Harmlessness, Logic, Objectivity, Relevance, Coherence, Length, Instruction Following, Completeness, Faithfulness, Result at the Beginning |
| category identification | Accuracy, Clarity, Relevance, Completeness, Faithfulness |
| title generation | Audience Friendly, Clarity, Creativity, Relevance, Engagement |
| question generation | Audience Friendly, Creativity, Harmlessness, Logic, Relevance, Attractive, Interactivity, Being Friendly, Coherence, Emojis, Emotion, Length, Style, Vivid |
| reading comprehension | Accuracy, Clarity, Depth, Insight, Completeness |
| keywords extraction | Accuracy, Clarity, Relevance, Completeness, Faithfulness |
| information extraction | Accuracy, Clarity, Relevance, Completeness, Faithfulness |
| topic modeling | Clarity, Information Richness, Relevance, Coherence, Completeness |
| ranking | Accuracy, Clarity, Relevance, Completeness, Faithfulness |
| data analysis | Audience Friendly, Creativity, Harmlessness, Logic, Relevance, Attractive, Interactivity, Being Friendly, Coherence, Emojis, Emotion, Length, Style, Vivid |
| **Aesthetic Needs** | |
| literary appreciation | Attractive, Being Friendly, Coherence, Coverage, Creativity, Depth, Harmlessness, Insight, Logic, Multiple Aspects, Originality, Professional, Relevance, Style, Vivid |
| language polish | Audience Friendly, Creativity, Depth, Harmlessness, Logic, Relevance, Attractive, Professional, Coherence, Style, Vivid, Instruction Following, Completeness, Faithfulness, Pointing Out |
| recommendation | Audience Friendly, Authenticity, Citation, Coverage, Depth, Harmlessness, Information Richness, Logic, Objectivity, Relevance, Timeliness, Interactivity, Being Friendly, Coherence, Instruction Following |
| creative writing | Audience Friendly, Creativity, Depth, Harmlessness, Information Richness, Insight, Logic, Multiple Aspects, Originality, Relevance, Attractive, Being Friendly, Coherence, Emotion, Style, Vivid, Instruction Following, Pacing |
| **Self-Actualization Needs** | |
| seeking advice | Accuracy, Admit Uncertainty, Audience Friendly, Authenticity, Citation, Clarity, Coverage, Feasibility, Harmlessness, Logic, Multiple Aspects, Relevance, Timeliness, Professional, Being Friendly, Completeness |
| planning | Accuracy, Audience Friendly, Clarity, Creativity, Feasibility, Harmlessness, Logic, Professionalism, Relevance, Timeliness, Interactivity, Instruction Following, Modularity, Completeness |
| brainstorming | Admit Uncertainty, Audience Friendly, Authenticity, Clarity, Coverage, Creativity, Depth, Feasibility, Harmlessness, Information Richness, Insight, Logic, Multiple Aspects, Objectivity, Originality, Relevance, Timeliness, Attractive, Interactivity, Professional |
| open question answering | Admit Uncertainty, Audience Friendly, Clarity, Creativity, Depth, Harmlessness, Information Richness, Insight, Logic, Multiple Aspects, Originality, Relevance, Being Friendly, Coherence, Style |
| value judgement | Admit Uncertainty, Audience Friendly, Authenticity, Coverage, Depth, Harmlessness, Insight, Logic, Multiple Aspects, Objectivity, Being Friendly |
| **Other** | |
| Other | Audience Friendly, Creativity, Harmlessness, Logic, Relevance, Attractive, Interactivity, Coherence, Emotion, Length, Style, Vivid |

## A2 DETAILS OF DATA ANNOTATION

In Section 3.3, we utilized GPT-4o to conduct fine-grained, dimension-level preference annotations. Specifically, using the prompts outlined in Table A3, we instructed GPT-4o to focus solely on the specified dimension, carefully compare the responses generated by two models, and select the better one or declare a tie. Additionally, we required GPT-4o to provide a pairwise comparison analysis, detailing the strengths and weaknesses of the two responses within the specified dimension to justify its decision.

Table A3: The prompt for GPT-4o fine-grained preference annotation.

[System Prompt]:

You are a helpful assistant. Given two dialogues with the same instruction, your task is to determine which dialogue is better. To fully utilize your capabilities, you will be provided with the scenario of the dialogues and a brief introduction to the scenario, as well as the attributes you need to consider. You first need to conduct a detailed comparison for each attribute, determining which dialogue is better for each attribute. Finally, please make a comprehensive judgment based on all the attributes to determine which dialogue is relatively better.

[User Prompt]:

Here are two dialogues with the same instruction. To avoid redundancy, we first provide the instruction of the dialogues, followed by the responses to the instruction.

[Instruction Start]:
{prompt}
[Instruction End]

[Response 1 Start]:
{response_a}
[Response 1 End]

[Response 2 Start]:
{response_b}
[Response 2 End]

To fully utilize your capabilities, you will be provided with the scenario of the dialogues and a brief introduction to the scenario, as well as the attributes you need to consider.

[Scenario Start]:
{scenario}
[Scenario End]

[Attributes Start]:
{attributes}
[Attributes End]

Please first conduct a detailed comparison for each attribute dependently (only analyze one attribute, you should ignore other attributes when comparing!), and then make a comprehensive judgment based on all the attributes to determine which dialogue is better. **You should weight the importance of each attribute, and guess the most important attributes that human user preferred (refer to the preference label).** You should extract detailed evidence when analyzing the performance of each dialogue. Your judgment results should only be '1' for dialogue 1 and '2' for dialogue 2. If their performances are similar, it should be 'tie'. Your output must follow the format below:

```
{{
  "Name of attribute 1": {{
    "comparison": "First extract detailed evidence, then fully analyze the performance of two dialogues on
        attribute 1 to make a comparison, no less than 50 words",
    "winner": "1, 2, or tie. 1 for dialogue 1 or 2 for dialogue 2, tie when their performances are similar"
  }},
  "Name of attribute 2": {{
    "comparison": "First extract detailed evidence, then fully analyze the performance of two dialogues on
        attribute 1 to make a comparison, no less than 50 words",
    "winner": "1, 2, or tie. 1 for dialogue 1 or 2 for dialogue 2, tie when their performances are similar"
  }},
  ...
  "Overall": {{
    "comparison": "First detailly analyze the most important attributes should be considered in this scenario
        (refer to the preference label) and give the weights of all attribute, finally make a comprehensive
        judgment based on weighted attributes, please analyze in detail.",
    "winner": "1, 2, or tie. 1 for dialogue 1 or 2 for dialogue 2, tie when their performances are similar"
  }}
}}
```

Overall winner should not be the dialogue that contains obvious deficiencies or defects. Ouput the above format in JSON. Again, the name of the attributes you should consider is {attributes_num}. You should ensure that your output include all the attributes. Do not output any other characters.

After collecting fine-grained preference annotations for all data, we employed a data pre-screening method inspired by STaR (Zelikman et al., 2022) to preliminarily enhance data quality. Specifically, we assumed that a reliable pairwise comparison analysis could effectively capture the strengths and weaknesses of the models, enabling GPT-4o to determine the correct overall winner by aggregating evaluations across all dimensions. This process is analogous to deriving an accurate answer through a reliable reasoning process (Zelikman et al., 2022).

To improve data quality, we compared the overall winners annotated by GPT-4o with the human-verified preference labels in the dataset. We retained 78% of the original data that matched human-verified preferences, thereby enhancing the dataset's alignment with human preferences.

## A3   QUALITY CONTROL

### A3.1   SCENARIO-SPECIFIC EVALUATION DIMENSIONS CONSTRUCTION

In Section 3.1, we mentioned that the evaluation dimensions required for each scenario were determined by three graduate students. To mitigate biases caused by the small sample size, we strictly adhered to the following measures to ensure the reliability of these decisions:

**Collecting Human Feedback.**   Before making any decisions, we encouraged the three graduate students to solicit feedback from volunteers by collecting 10 questionnaires for each scenario. These questionnaires aimed to investigate which dimensions humans tend to consider in each specific scenario. The graduate students selected dimensions with a selection rate exceeding 60% in each scenario as candidate dimensions for further discussion.

**Referencing Related Works.**   We referred to related studies, such as Auto-J, which provided insights into dimension selection for specific scenarios. While these studies differ from our work in terms of scenario design, they offered valuable inspiration for dimension selection in similar contexts.

Based on these efforts, the final decisions were collectively made through discussions among the three graduate students. Therefore, although some dimensions may have been overlooked, the retained dimensions for each scenario were carefully reviewed and scrutinized to ensure their validity.

### A3.2   HUMAN VERIFICATION OF MD-EVAL

We detail the process of constructing MD-Eval as follows:

**Data Source**   MD-Eval data was derived from a subset of the data constructed in Section 3.1-3.3, with 10 samples collected for each scenario, resulting in a total of 360 samples ($5,482$ dimension-level annotations). It is important to note that while using GPT-4o for data annotation, we additionally generated pairwise comparison analyses for each dimension to serve as references for subsequent human annotators.

**Human Verification Process**   We enlisted three graduate students with foundational NLP research experience to verify the data. Specifically, the participants reviewed GPT-4o's analysis for each dimension to identify any obvious biases and then provided validated preference annotations. We ensured that these participants annotated every dimension in each sample.

**Final Annotation Agreement**   The final preference label for each dimension was determined by the majority vote among participants. We defined the inter-annotator agreement as the average proportion of participants selecting the most chosen preference for each dimension. The agreement among the annotators was calculated to be $87.61\%$.

**Comparison with GPT-4o Annotations**   We compared the human-validated labels with GPT-4o annotations and found an agreement of $82.49\%$. This demonstrates that the annotation process described in Section 3.1-3.3 resulted in GPT-4o labels that align closely with human annotations and exhibit a level of agreement comparable to that among human annotators, further supporting the reliability of the training data.

## A4 ADDITIONAL DISCUSSIONS

### A4.1 ABLATION STUDY ON $\lambda_1$ AND $\lambda_2$

As presented in Table A4, we explored the impact of multiple combinations of $\lambda_1$ and $\lambda_2$ in Eq. (4) by systematically increasing and decreasing their values. When $\lambda_1 > \lambda_2$ (e.g., $\lambda_1 = 1.5$, $\lambda_2 = 1$), we observed a performance improvement in the pairwise comparison task but a slight degradation in the single rating task. This can be attributed to the fact that in the pairwise comparison setting, SaMer evaluates two responses using the same evaluation dimensions and their associated weights, making dimension-level scoring capabilities particularly critical. Since $\lambda_1$ controls the rank loss at the dimension level, it plays a direct role in this context. Conversely,

Table A4: Impact of $\lambda_1$ and $\lambda_2$ on SaMer performance in single rating and pairwise comparison tasks.

| $\lambda_1, \lambda_2$ | Vicuna Bench (%) | Auto-J Eval (%) | |
|---|---|---|---|
| | Pearson | w/o TIE | w/ TIE |
| $1, 1$ | 47.59 | 76.15 | 57.61 |
| $0.5, 1$ | 47.15 | 75.86 | 57.33 |
| $1.5, 1$ | 47.58 | 76.25 | 57.76 |
| $1, 0.5$ | 47.39 | 76.35 | 57.76 |
| $1, 1.5$ | 47.97 | 75.37 | 56.97 |

in the single rating setting, accurately predicting both evaluation dimensions and their weights becomes essential, and excessively low $\lambda_2$ values hinder the accurate prediction of these weights.

When $\lambda_1 < \lambda_2$ (e.g., $\lambda_1 = 1$, $\lambda_2 = 1.5$), we found that SaMer's performance improved in the single rating task but deteriorated in the pairwise comparison task. This indicates that increasing $\lambda_2$ indeed benefits the prediction of dimension weights, thereby enhancing the accuracy of single rating tasks, further validating our earlier analysis. However, in the case of $\lambda_1 = 0.5$ and $\lambda_2 = 1$, both single rating and pairwise comparison tasks exhibited poor performance, underscoring the critical importance of dimension-level scoring (i.e., $\lambda_1$).

In summary, considering the respective roles and significance of $\lambda_1$ and $\lambda_2$, our 1design choice of setting $\lambda_1 = \lambda_2 = 1$ is both reasonable and effective.

### A4.2 INFERENCE EFFICIENCY OF SAMER

We further evaluate the inference efficiency of SaMer compared to other large language models with a similar parameter scale (i.e., 7–8B). Experiments were conducted using an NVIDIA GeForce RTX 4090 GPU, with all model parameters stored in bf16 precision. The inference framework employed is based on HuggingFace `transformers` Python library (Wolf, 2020).

As shown in Table A5, SaMer exhibits a significantly lower average runtime for processing a single data instance compared to the other baselines, highlighting its superior efficiency. This efficiency can be attributed to SaMer's design as a classification model, which requires

Table A5: Inference efficiency comparison of SaMer and baseline LLMs (*average runtime in **seconds** per instance*).

| **Evaluator** | Single Rating ($\downarrow$) | Pairwise Comparison ($\downarrow$) |
|---|---|---|
| Llama2-7B-Chat | 0.29 | 0.23 |
| Llama3-8B-Inst | 0.11 | 0.28 |
| Llama3.1-8B-Inst | 0.10 | 0.19 |
| Promethus-7B | 0.74 | 1.26 |
| Promethus2-7B | 0.64 | 0.57 |
| **SaMer-8B** | **0.04** | **0.10** |

only a single forward pass through MLP layers to generate the final classification result. In contrast, the baselines, being causal language models, involve the generation and analysis of text segments during evaluation, requiring multiple forward passes, which ultimately reduces their inference speed.

### A4.3 EXPLANATION OF SAMER'S OUTSTANDING PERFORMANCE IN SINGLE RATING

In Table 2, the Pearson correlation of SaMer on the Vicuna Bench (Chiang et al., 2023) is comparable to proprietary models, i.e., GPT-4o/4o-mini (OpenAI, 2024a) and Claude-3.5-Sonnet (Anthropic, 2024), while its Spearman and Kendall correlations are significantly higher. This observation suggests that, despite GPT-4o performing all annotations for the preference dataset construction (in 3), SaMer outperforms GPT-4o on these metrics. Furthermore, the substantial difference between the Spearman and Kendall correlations and the Pearson correlation warrants further explanation.

We summarize three key factors to explain why SaMer outperforms GPT-4o in single-rating tasks:

1. **SaMer employs the preference data for training:** It's important to emphasize that the data used to train SaMer was not derived from GPT-4o's single ratings. Instead, when constructing the preference data for SaMer's training, we instructed GPT-4o to perform pair-

wise comparisons, selecting the better response between two options. For LLMs, pairwise comparison is considered a more reliable evaluation method than single rating (Liu et al., 2024). Therefore, it is possible that SaMer, trained on preference data, could outperform GPT-4o in single-rating evaluations.

2. **Fine-grained evaluation provides an advantage over coarse-grained evaluation:** When using GPT-4o for preference annotation, we focused on single-dimension preferences, allowing SaMer to integrate preference information across multiple dimensions to compute an aggregated score. In contrast, when evaluating GPT-4o on Vicuna Bench, it was only asked to perform coarse-grained scoring.

3. **Differences in scoring range and score discreteness:** In typical LLM judging scenarios on Vicuna Bench, scoring is usually done on a discrete scale of 1–5 (similar to a classification task with five categories). SaMer, however, outputs a continuous score within the range (0, 1), making it more sensitive to quality variations.

Additionally, Spearman correlation is rank-based, measuring the consistency of rank order between two variables, while Kendall correlation compares pairwise order consistency between samples. **These metrics are better suited for handling continuous, non-repeated values.** When duplicate values are present, as often occurs in the Vicuna Bench where GPT-4o frequently assigns the same score (e.g., 232 out of 320 responses received a score of 5), the rank information becomes incomplete, negatively impacting these correlations. In contrast, SaMer produces fewer duplicate scores, maintaining better rank-order integrity.

