# OpenReview forum: "SaMer: A Scenario-aware Multi-dimensional Evaluator for Large Language Models"
_ICLR.cc/2025/Conference — ICLR 2025 Poster_

### Official Review · Reviewer_DeNC · 2024-11-04

**Soundness:** 2
**Presentation:** 3
**Contribution:** 3
**Rating:** 6
**Confidence:** 4

**Summary:**

This paper presents SaMer, a new framework to evaluate LLM responses with more flexibility. Instead of using static criteria for evaluating open-ended generation, SaMer adapts its approach based on the context, whether it's looking at a creative response, answering a factual question, or offering advice. To achieve this, the authors created a large-scale dataset covering a variety of real-life scenarios so SaMer could learn to recognize what matters most in different situations. The framework is set up with three parts: one to pick relevant criteria, one to score based on those criteria, and one to weigh them for an overall score.

Experiments indicate that SaMer performs competitively, achieving improvements over baseline models across single rating, pairwise comparison, and fine-grained dimension-specific benchmarks. It also sheds light on which factors are key in different scenarios. Overall, SaMer brings a more adaptable, context-sensitive way to evaluate LLM responses, making it better suited to assess responses based on what's actually needed in each situation.

**Strengths:**

1. The motivation for a more fine-grained, multi-dimensional evaluation of LLM responses is reasonable and relatively under-explored, as current evaluations usually rely on a single overall score.
2. The authors have built a large-scale preference dataset from existing corpora, covering diverse scenarios and annotating each with corresponding evaluation dimensions.
3. Experimentally, the model shows performance improvements over baselines across several settings.

**Weaknesses:**

**Major Issues:**

1. **The choice of evaluation dimensions for each scenario is unclear**: The authors mention that three graduate students annotated the evaluation dimensions for each scenario. However, this task seems highly subjective and challenging. From Table A2, many dimensions appear vague or redundant. For instance, why doesn’t "data analysis" include "Accuracy"? Why is "Logic" missing for "reading comprehension"? “Recommendation” seems subjective, so why does it include "Objectivity"? "Accuracy" and "Admit Uncertainty" also seem somewhat contradictory, yet both are included in five scenarios. "Being Friendly" and "Audience Friendly" seem overlapping. Overall, the annotations lack clarity and intuitiveness. The authors could improve this by defining these dimensions and including the guidelines used for annotation in the paper. Additionally, reporting inter-annotator agreement with discussion would be helpful.

2. **The predicted evaluation dimensions are still pre-defined**: Although SaMer claims to flexibly predict relevant evaluation dimensions per scenario, it actually selects from a set of 42 dimensions defined in previous studies. This limits SaMer's ability to generalize to broader, unforeseen scenarios.

3. **No evaluation of selected evaluation dimensions**: The experiments in this paper focus on overall response preference or specific fixed dimensions. However, one of SaMer's primary contributions is automating the prediction of relevant dimensions for each scenario, yet there is no experiment supporting SaMer’s effectiveness in this. A potential experiment could involve having LLMs generate reasonable evaluation dimensions for given scenarios as a baseline, then comparing those results with SaMer's using human judgment as a metric.

4. **Missing related work**: Multi-dimensional and explainable evaluations for text generation are not new ideas, and it would be helpful for the authors to include comparisons and discussions with prior works such as [1-3].

**Minor Issues:**

1. **Potential unfair comparisons**: The training of SaMer heavily relies on GPT-4o for preference annotation, while some baselines may not have used this newer and potentially stronger teacher model, which could lead to unfair comparisons.
2. **Lack of efficiency discussion**: Since SaMer first predicts evaluation dimensions before aggregating them into an overall score, discussing runtime and efficiency comparisons would help the community better understand the practical efficiency of this framework.

**References:**

[1] Zhong et al. Towards a Unified Multi-Dimensional Evaluator for Text Generation. EMNLP 2022.

[2] Liu et al. G-Eval: NLG Evaluation using GPT-4 with Better Human Alignment. EMNLP 2023.

[3] Xu et al. INSTRUCTSCORE: Explainable Text Generation Evaluation with Finegrained Feedback. EMNLP 2023.

**Questions:**

In Table 2, SaMer's Pearson correlation on the Vicuna Bench is comparable to GPT-4o, but its Spearman and Kendall correlations are significantly higher. Given that GPT-4o performed all preference annotations, why does SaMer outperform GPT-4o on these metrics, and are there any insights into why these correlations differ so much from the Pearson correlation?

---

> ### Author Response · Authors · 2024-11-21
> **Response to Reviewer DeNC (1)**
>
> Thanks for your comments!
> > The authors mention that three graduate students annotated the evaluation dimensions for each scenario. However, this task seems highly subjective and challenging. From Table A2, many dimensions appear vague or redundant. For instance, why doesn’t "data analysis" include "Accuracy"? Why is "Logic" missing for "reading comprehension"? “Recommendation” seems subjective, so why does it include "Objectivity"? "Accuracy" and "Admit Uncertainty" also seem somewhat contradictory, yet both are included in five scenarios. "Being Friendly" and "Audience Friendly" seem overlapping. Overall, the annotations lack clarity and intuitiveness. The authors could improve this by defining these dimensions and including the guidelines used for annotation in the paper. Additionally, reporting inter-annotator agreement with discussion would be helpful.
>
> We apologize for not including a detailed discussion on dimension definitions and annotation guidelines in the appendix, which may have caused some confusion. In fact, defining the set of dimensions and selecting appropriate dimensions for each scenario was the first and one of the most critical steps in our work.
>
> As mentioned in Section 3.1 (lines 153-154), the majority of our 42 dimensions were collated and summarized from prior research, providing a comprehensive coverage of dimensions considered in existing studies. We have supplemented the definitions of these dimensions in each scenario at (https://anonymous.4open.science/r/SaMer-2878/utils/prompt/metrics.yaml). While the final decisions regarding the evaluation dimensions were made by three graduate students, we took the following measures to ensure the reliability of these decisions:
>
> 1. **Human Feedback Collection**: Before making any decisions, we encouraged the three graduate students to seek input from volunteers by collecting 10 questionnaires for each scenario. These questionnaires aimed to investigate which dimensions humans preferred to consider for each specific scenario. The graduate students selected dimensions with a choice rate exceeding 60% in each scenario as candidate dimensions for further discussion.
> 2. **Reference to Related Work**: We referred to related studies, such as Auto-J [1], which provided insights on dimension selection for specific scenarios. Although these studies had some differences in scenario design compared to our work, they offered valuable inspiration for dimension selection in similar contexts.
>
> Building on these efforts, the final decisions were made collaboratively through discussion. Regarding your concerns about missing or overlapping dimensions, we acknowledge that some gaps may exist. However, the dimensions retained for each scenario were carefully reviewed and checked to ensure their reasonableness.
>
> [1] Li J, Sun S, Yuan W, et al. Generative Judge for Evaluating Alignment. ICLR2024.

---

> > ### Author Response · Authors · 2024-11-21
> > **Response to Reviewer DeNC (4)**
> >
> > Thanks for your comments!
> > > Potential unfair comparisons: The training of SaMer heavily relies on GPT-4o for preference annotation, while some baselines may not have used this newer and potentially stronger teacher model, which could lead to unfair comparisons.
> >
> > We acknowledge your concern regarding the potential unfairness of comparisons. This issue may persist over time, primarily due to the short lifespan of LLM versions. For instance, Prometheus uses GPT-4 for data annotation, while Prometheus 2, as its subsequent iteration, uses the GPT-4-1106-preview model for annotation. On one hand, we believe that advancements in methodology improve model capabilities; on the other hand, improvements in teacher models result in higher-quality data annotations, which also enhance model performance. We look forward to related work continually updating their training data (using the latest models for annotation) to facilitate more fair comparisons in the future.
> >
> > ---
> >
> > > Lack of efficiency discussion: Since SaMer first predicts evaluation dimensions before aggregating them into an overall score, discussing runtime and efficiency comparisons would help the community better understand the practical efficiency of this framework.
> >
> > Thank you for your attention to SaMer's inference efficiency. To further clarify the inference efficiency of SaMer compared to other LLMs of a similar parameter scale (i.e., 7–8B), we conducted additional experiments, as shown in Table R3. In these experiments, we used an NVIDIA GeForce RTX 4090 GPU, with the inference framework being `transformers` and all model parameters in bf16 precision.
> >
> > As demonstrated in Table R3, SaMer’s average runtime (s) for processing a single data instance is significantly lower than that of other baselines, validating SaMer's efficiency. This advantage is attributed to SaMer’s nature as a classification model, requiring only a single forward pass through an MLP layer to produce the final classification result. In contrast, the baselines are causal language models, which typically involve generating and analyzing a segment of text during evaluation, necessitating multiple forward passes and thus slowing down the inference speed.
> >
> > **Table R3**. Inference Efficiency Comparison of SaMer and Baseline LLMs  (_Average Runtime in **Seconds** per Instance_).
> >
> > |Model|Single Rating|Pairwise Comparison|
> > |-----|-------------|--------------------|
> > |Llama2-7B-Chat|0.29|0.23|
> > |Llama3-8B-Inst|0.11|0.28|
> > |Llama3.1-8B-Inst|0.10|0.19|
> > |Promethus-7B|0.74|1.26|
> > |Promethus2-7B|0.64|0.57|
> > |SaMer-8B|0.04|0.10|

---

> ### Author Response · Authors · 2024-11-21
> **Response to Reviewer DeNC (2)**
>
> Thanks for your comments!
> > No evaluation of selected evaluation dimensions: The experiments in this paper focus on overall response preference or specific fixed dimensions. However, one of SaMer's primary contributions is automating the prediction of relevant dimensions for each scenario, yet there is no experiment supporting SaMer’s effectiveness in this. A potential experiment could involve having LLMs generate reasonable evaluation dimensions for given scenarios as a baseline, then comparing those results with SaMer's using human judgment as a metric.
>
> Thank you for your insightful suggestion! To better demonstrate SaMer's accuracy in dimension prediction, **we compared SaMer with generative LLM baselines on the task of dimension selection**. To ensure consistency in the dimension space, we used the 42 dimensions mentioned in the paper as the overall set. For the baselines, we guided them via prompts to select the most suitable subset of dimensions from this set for each scenario. SaMer’s performance was evaluated on both **in-domain** (ID) and **out-of-domain** (OOD) datasets, as shown in Table R1. The in-domain test set was MD-Eval, while the OOD test set was Auto-J Eval. For Auto-J Eval, we manually filtered out scenarios overlapping with the 36 scenarios in our dataset, resulting in 108 unique examples (details available at https://anonymous.4open.science/r/SaMer-2878/data/benchmark/OOD/AutoJ_Eval.json).
>
> For the ID evaluation, we employed the Precision and Recall as metric. For the OOD evaluation, as there is no ground truth for dimension selection, we calculated the win rate against SaMer (ie, the proportion of cases where a baseline outperformed SaMer according to human judgment).
>
> The results (Table R1) show that SaMer outperms baselines. **In the in-domain evaluation**, the precision of most baselines does not exceed 50%, and recall remains below 40%. This indicates that current LLMs are not yet capable of effectively addressing both challenges. Among the baselines, GPT-4o demonstrates relatively better performance, with high precision but low recall, suggesting that its selected subset of dimensions is smaller (less comprehensive) but highly accurate. **In the out-of-domain evaluation**, GPT-4o’s win rate does not exceed 50%, highlighting that SaMer’s dimension selection aligns more closely with human preferences, even surpassing proprietary models in this regard.
>
> In conclusion, selecting appropriate dimensions from the 42-dimension space is challenging, requiring a balance between coverage and precision. While some baselines demonstrated partial strengths (e.g., GPT-4o's high precision), none achieved the overall balance that SaMer provides. This experiment supports SaMer's ability to effectively and flexibly predict evaluation dimensions across diverse scenarios.
>
> We appreciate your suggestion to include such an evaluation, as it highlights SaMer's core strength and aligns well with its intended contributions.
>
> **Table R1**. Performance Comparison of SaMer and LLM Baselines on Dimension Selection Tasks (ID and OOD).
>
> |Model|ID Precision|ID Recall|OOD Win Rate (against SaMer)|
> |-----|------------|---------|---------------------------|
> |GPT-4o|63.42|38.10|48.15|
> |GPT-4o-mini|57.61|37.89|37.65|
> |Mistral-7B-Inst|42.73|23.82|14.42|
> |Llama2-7B-Chat|27.78|37.36|10.84|
> |Llama3-8B-Inst|39.81|36.37|21.61|
> |Llama3.1-8B-Inst|34.96|60.55|28.06|
> |SaMer|74.84|72.33|-|

---

> ### Author Response · Authors · 2024-11-21
> **Response to Reviewer DeNC (3)**
>
> Thanks for your comments!
>
> > The predicted evaluation dimensions are still pre-defined: Although SaMer claims to flexibly predict relevant evaluation dimensions per scenario, it actually selects from a set of 42 dimensions defined in previous studies. This limits SaMer's ability to generalize to broader, unforeseen scenarios.
>
> We appreciate your concern regarding the limitation of pre-defined evaluation dimensions. We have made every effort to curate and collect a comprehensive set of dimensions. While we acknowledge that some important dimensions may still be missing, to the best of our knowledge, the multi-dimensional evaluator proposed in our work already includes one of the largest sets of dimensions among existing models. We believe that most scenarios can find suitable subsets of dimensions for evaluation from our collection.
>
> To further enable SaMer to flexibly select dimensions for each scenario, we defined 36 common scenarios (Fig. 2) as demonstrations and trained SaMer to predict and assign unique dimensions to different scenarios. During inference, given a specific prompt, SaMer leverages its "dimension prediction layer" to identify the corresponding scenario and assign relevant dimensions. Importantly, the scenarios during inference are not necessarily restricted to the predefined 36 scenarios but may include novel, undefined scenarios. Since any new scenario will likely share similarities with one or more of the predefined scenarios, SaMer can infer and adapt dimensions for new scenarios based on similar predefined ones. For this reason, we believe "**SaMer can flexibly predict relevant evaluation dimensions per scenario**".Details of the experiments evaluating SaMer's dimensional prediction capability are provided in **Table R1**.
>
> However, we also recognize that if a novel scenario requires key evaluation dimensions entirely unrelated to the existing set of 42, SaMer might indeed fail. To address this limitation, we plan to continuously expand the dimension set and predefined scenarios to further enhance SaMer's flexibility and generalizability.
>
> ---
>
> > Multi-dimensional and explainable evaluations for text generation are not new ideas, and it would be helpful for the authors to include comparisons and discussions with prior works such as [1-3].
>
> The idea of multi-dimensional and explainable evaluations for text generation is not novel, as highlighted by prior works [1-3]. Here, we emphasize the differences between our motivation and approach compared to the existing works [1-3]:
>
> 1. The UNIEVAL and G-EVAL models proposed in [1][2] are fine-grained dimension-level evaluators, capable of providing specific scores for individual dimensions. This is similar to the "Scoring Layer" in SaMer, which scores individual dimensions. However, these models do not integrate all dimensions to provide an instance-level overall score. Additionally, **these models are configured with only a small and fixed set of evaluation dimensions**. Compared to SaMer, their evaluation results are overly specific and fragmented.
>
>     Additionally, one of the key motivations behind SaMer is to adaptively design different weights for dimensions based on the scenario. For example, "harmlessness" is critical in safety evaluation scenarios but less relevant in mathematical contexts. Finally, our 42 defined dimensions are more comprehensive and detailed, offering broader coverage compared to the existing works.
>
> 2. INSTRUCTSCORE, proposed in [3], is an evaluation framework focused on error detection, capable of generating error types, error locations, severity levels, and corresponding explanations. It also includes 10 failure modes to ensure evaluation accuracy. However, compared to SaMer, **INSTRUCTSCORE primarily focuses on error diagnosis and lacks a multi-dimensional focus on positive attributes of generated text, leading to biased evaluation results**. Furthermore, it lacks a weight adjustment mechanism to integrate different error types (or dimensions) into a comprehensive score, making it less flexible.
>
> In conclusion, while we appreciate the contributions of previous works to multi-dimensional and explainable evaluations, the core contributions of SaMer lie in its scenario-aware adaptive dimension design and support for flexible evaluation modes. Moreover, the extensive set of evaluation dimensions makes SaMer’s evaluation more fine-grained.
>
> In the revised manuscript, we will provide additional discussion on the prior works.

---

> ### Author Response · Authors · 2024-11-21
> **Response to Reviewer DeNC (5)**
>
> Thanks for your comment!
>
> > In Table 2, SaMer's Pearson correlation on the Vicuna Bench is comparable to GPT-4o, but its Spearman and Kendall correlations are significantly higher. Given that GPT-4o performed all preference annotations, why does SaMer outperform GPT-4o on these metrics, and are there any insights into why these correlations differ so much from the Pearson correlation?
>
> We summarize three key factors to explain why SaMer outperforms GPT-4o in **single-rating** tasks (Table 2):
>
> 1. **SaMer employs the preference data for training**: It's important to emphasize that the data used to train SaMer was not derived from GPT-4o’s single ratings. Instead, when constructing the preference data for SaMer's training, we instructed GPT-4o to perform pairwise comparisons, selecting the better response between two options. For LLMs, pairwise comparison is considered a more reliable evaluation method than single rating [1]. Therefore, it is possible that SaMer, trained on preference data, could outperform GPT-4o in single-rating evaluations.
> 2. **Fine-grained evaluation provides an advantage over coarse-grained evaluation**: When using GPT-4o for preference annotation, we focused on single-dimension preferences, allowing SaMer to integrate preference information across multiple dimensions to compute an aggregated score. In contrast, when evaluating GPT-4o on Vicuna Bench, it was only asked to perform coarse-grained scoring.
> 3. **Differences in scoring range and score discreteness**: In typical LLM judging scenarios on Vicuna Bench, scoring is usually done on a discrete scale of 1–5 (similar to a classification task with five categories). SaMer, however, outputs a continuous score within the range (0, 1), making it more sensitive to quality variations.
>
> Additionally, Spearman correlation is rank-based, measuring the consistency of rank order between two variables, while Kendall correlation compares pairwise order consistency between samples. These metrics are better suited for handling continuous, non-repeated values. When duplicate values are present, as often occurs in the Vicuna Bench where GPT-4o frequently assigns the same score (e.g., 232 out of 320 responses received a score of 5), the rank information becomes incomplete, negatively impacting these correlations. In contrast, SaMer produces fewer duplicate scores, maintaining better rank-order integrity.
>
> Finally, we provide a link (https://anonymous.4open.science/r/SaMer-2878/README.md, Tips 5) to the actual experimental results and the corresponding metrics calculation process to ensure transparency and authenticity.
>
> [1] Liu Y, Zhou H, Guo Z, et al. Aligning with human judgement: The role of pairwise preference in large language model evaluators. arXiv:2403.16950, 2024.

---

> ### Author Response · Authors · 2024-11-25
>
> We sincerely appreciate the time and effort you have dedicated to reviewing our manuscript and providing valuable suggestions! As the author-reviewer discussion phase draws to a close, we would like to confirm whether our responses have effectively addressed your concerns. A few days ago, we provided detailed replies to your comments, and we hope they have adequately resolved the issues you raised. Should you require further clarification or have any additional questions, please do not hesitate to reach out to us. We are more than willing to continue the communication with you. Thank you once again, and we look forward to your further feedback!

---

> ### Comment · Reviewer_DeNC · 2024-11-25
> **Thank you for your response**
>
> Thank you for the detailed rebuttal and clarifications. They have indeed addressed my concerns regarding:
> - Lack of experiments for the selected evaluation dimension
> - Absence of discussion on related works for multi-dimensional evaluators
> - Absence of discussions on efficiency
>
> As a result, I have decided to increase my score from 5 to 6.
>
> That said, there are still some factors that, in my view, prevent this submission from achieving a higher score:
>
> - While I appreciate the authors' effort in the annotations, I still find Table A2 insufficiently intuitive and unconvincing
> - The evaluation dimensions remain heavily reliant on manually defined categories, which are limited to 42 dimensions
> - As the authors themselves acknowledge, the rapid development of LLMs leads to potentially unfair comparisons
>
> In summary, I hold a slightly positive opinion of this submission and strongly encourage the authors to incorporate all additional experiments and discussions mentioned in their responses into the next version of the paper, especially those concerning the experiments for the selected evaluation dimensions.

---

### Official Review · Reviewer_EYLr · 2024-11-04

**Soundness:** 2
**Presentation:** 3
**Contribution:** 2
**Rating:** 6
**Confidence:** 4

**Summary:**

This paper proposes an evaluation model (SaMer) that assess the quality of responses based a 42 pre-selected dimensions and gives an overall evaluation by weighting scores on different dimensions. The experimental results show that the proposed model achieves the best efficacy on single rating datasets and the fine-grained comparion data curated by the authors, and it also achieves a comparable performance with ArmoRM on pairwise comparison datasets.

**Strengths:**

- Compared to past evaluators that assessed overall quality, the fine-grained evaluation method proposed in this paper provides more fine-grained evaluation results. The weights across different dimensions offer a certain level of interpretability.

- Since this model can implicitly determine the required dimensions based solely on the query without task information, this evaluator is task-agnostic and possesses a certain degree of generality.

**Weaknesses:**

- The evaluator's ability to understand different dimensions is questionable. Firstly, the 42 dimensions are fixed, and the model did not even encounter these dimension definitions during training (nor did the paper provide their specific definitions). Secondly, the experimental results do not demonstrate whether the evaluator can accurately determine which dimensions are needed when encountering out-of-domain data. Compared to similar work like X-Eval [1], which allows users to input personalized dimension definitions, it is hard to say that this model has a clear advantage.

- The paper employs three loss functions but lacks ablation studies to investigate the impact of choosing $\lambda_1$ and $\lambda_2$ on model performance, making it difficult for readers to understand the individual contributions of these designs.



[1] X-Eval: Generalizable Multi-aspect Text Evaluation via Augmented Instruction Tuning with Auxiliary Evaluation Aspects (Liu et al., NAACL 2024)

**Questions:**

- Typos: The term "Demension" in Figure 3 should be corrected to "Dimension."

- How is the dimension-level accuracy in Table 5 calculated?

- The data used to train fine-grained preferences is primarily annotated by GPT-4o; however, GPT-4o has issues with confusing different dimensions [2]. How can you ensure that the data generated by GPT-4o is of high quality?

- Could you please provide more information on how you verify the quality of MD-Eval test set (used in Table 5) through human evaluation?

[2] Are LLM-based Evaluators Confusing NLG Quality Criteria? (Hu et al., ACL 2024)

---

> ### Author Response · Authors · 2024-11-21
> **Response to Reviewer EYLr (1)**
>
> Thanks for your comment!
> > The evaluator's ability to understand different dimensions is questionable. Firstly, the 42 dimensions are fixed, and the model did not even encounter these dimension definitions during training (nor did the paper provide their specific definitions). Secondly, the experimental results do not demonstrate whether the evaluator can accurately determine which dimensions are needed when encountering out-of-domain data. Compared to similar work like X-Eval [1], which allows users to input personalized dimension definitions, it is hard to say that this model has a clear advantage.
>
> We apologize for any confusion caused by not including the definitions of the 42 dimensions in the appendix. The precise descriptions of each dimension for every scenario are available at https://anonymous.4open.science/r/SaMer-2878/utils/prompt/metrics.yaml. In fact, these definitions were used during the annotation of preference data. Specifically, we instructed GPT-4o to perform dimension-level pairwise comparisons based on the detailed definitions of each dimension. During SaMer training, the dimension prediction task was modeled as a multi-label classification problem, so incorporating these explicit definitions was unnecessary.
>
> To demonstrate SaMer's accuracy in dimension prediction, we compared its performance with that of generative LLM baselines on dimension selection task. To ensure consistency in the dimension space, the full set of dimensions was fixed to the 42 mentioned in the paper. For the baselines, we used prompts to guide them in selecting the most suitable subset of dimensions for a given scenario from the 42-dimension set. SaMer’s performance was evaluated on both in-domain (ID) and out-of-domain (OOD) datasets, as shown in Table R1. The in-domain test set was MD-Eval, while the OOD test set was Auto-J Eval. For Auto-J Eval, we manually filtered out scenarios overlapping with the 36 scenarios in our dataset, resulting in 108 unique examples (details available at https://anonymous.4open.science/r/SaMer-2878/data/benchmark/OOD/AutoJ_Eval.json).
>
> For the ID evaluation, we employed the Precision and Recall as metric. For the OOD evaluation, as there is no ground truth for dimension selection, we calculated the win rate against SaMer (ie, the proportion of cases where a baseline outperformed SaMer according to human judgment).
>
> The results （Table R1) show that SaMer outperms baselines. **In the in-domain evaluation**, the precision of most baselines does not exceed 50%, and recall remains below 40%. This indicates that current LLMs are not yet capable of effectively addressing both challenges. Among the baselines, GPT-4o demonstrates relatively better performance, with high precision but low recall, suggesting that its selected subset of dimensions is smaller (less comprehensive) but highly accurate. **In the out-of-domain evaluation**, GPT-4o’s win rate does not exceed 50%, highlighting that SaMer’s dimension selection aligns more closely with human preferences, even surpassing proprietary models in this regard.
>
>
> Finally, regarding your mention of related work, **X-Eval**, while it allows for personalized dimension definitions, it still requires **a fixed set of "auxiliary aspects" to be defined first**—similar to the predefined dimension set used in SaMer. During inference, X-Eval selects the top-k (with **k=1** in their experiments) auxiliary aspects most semantically similar to the target dimension and evaluates these aspects as cues for the target dimension.
>
> Moreover, in contrast to SaMer, X-Eval is designed for evaluating Boolean QA tasks on a specific target dimension (e.g., "Is this sentence informative according to the reference?") and is **not suitable for single-score or pairwise comparison tasks**. We emphasize that SaMer’s strength lies in its ability to adaptively identify relevant dimensions for different scenarios, integrate scores across these dimensions, and produce an overall response score. This flexibility makes it more suitable for the tasks considered in our work.
>
> **Table R1**. Performance Comparison of SaMer and LLM Baselines on Dimension Selection Tasks (ID and OOD)
>
> |Model|ID Precision|ID Recall|OOD Win Rate (against SaMer)|
> |-----|------------|---------|---------------------------|
> |GPT-4o|63.42|38.10|48.15|
> |GPT-4o-mini|57.61|37.89|37.65|
> |Mistral-7B-Inst|42.73|23.82|14.42|
> |Llama2-7B-Chat|27.78|37.36|10.84|
> |Llama3-8B-Inst|39.81|36.37|21.61|
> |Llama3.1-8B-Inst|34.96|60.55|28.06|
> |SaMer|74.84|72.33|-|

---

> ### Author Response · Authors · 2024-11-21
> **Response to Reviewer EYLr (2)**
>
> Thanks for your comments!
>
> > The paper employs three loss functions but lacks ablation studies to investigate the impact of choosing λ1 and λ2 on model performance, making it difficult for readers to understand the individual contributions of these designs.
> Thank you very much for your valuable suggestions! To further analyze the robustness of the SaMer training method, we investigated the model's performance under different values of λ1 and λ2 in the single rating and pairwise comparison evaluation modes, as shown in Table R2. Specifically, we utilized Vicuna Bench to evaluate the performance on the single rating task and Auto-J Eval to assess the performance on the pairwise comparison task.
>
> As presented in Table R2, we explored the impact of multiple combinations of λ1 and λ2 by systematically increasing and decreasing their values. When λ1 > λ2 (e.g., λ1 = 1.5, λ2 = 1), we observed a performance improvement in the pairwise comparison task but a slight degradation in the single rating task. This can be attributed to the fact that in the pairwise comparison setting, SaMer evaluates two responses using the same evaluation dimensions and their associated weights, making dimension-level scoring capabilities particularly critical. Since λ1 controls the rank loss at the dimension level, it plays a direct role in this context. Conversely, in the single rating setting, accurately predicting both evaluation dimensions and their weights becomes essential, and excessively low λ2 values hinder the accurate prediction of these weights.
>
> When λ1 < λ2 (e.g., λ1 = 1, λ2 = 1.5), we found that SaMer's performance improved in the single rating task but deteriorated in the pairwise comparison task. This indicates that increasing λ2 indeed benefits the prediction of dimension weights, thereby enhancing the accuracy of single rating tasks, further validating our earlier analysis. However, in the case of λ1 = 0.5 and λ2 = 1, both single rating and pairwise comparison tasks exhibited poor performance, underscoring the critical importance of dimension-level scoring (i.e., λ1).
>
> In summary, considering the respective roles and significance of λ1 and λ2, our 1design choice of setting λ1 = λ2 = 1 is both reasonable and effective.
>
> **Table R2**. Impact of λ1 and λ2 on SaMer Performance in Single Rating and Pairwise Comparison Tasks.
> |λ1,λ2|Vicuna Bench|Auto-J Eval (w/o tie)|Auto-J Eval (w/ tie)|
> |-----|------------|---------------------|--------------------|
> |1,1(default)|47.59|76.15|57.61|
> |0.5,1|47.15|75.86|57.33|
> |1.5,1|47.58|76.25|57.76|
> |1,0.5|47.39|76.35|57.76|
> |1,1.5|47.97|75.37|56.97|
>
> ---
>
> > How is the dimension-level accuracy in Table 5 calculated?
>
> We first require the baselines to perform pairwise comparisons for each response pair on individual dimensions, generating dimension-level preference predictions. Finally, the dimension-level accuracy is calculated as the average accuracy of these predictions against the ground truth preference labels.

---

> ### Author Response · Authors · 2024-11-21
> **Response to Reviewer EYLr (3)**
>
> Thanks for your comments!
> > The data used to train fine-grained preferences is primarily annotated by GPT-4o; however, GPT-4o has issues with confusing different dimensions [2]. How can you ensure that the data generated by GPT-4o is of high quality?
>
> We understand your concerns regarding the potential issues with GPT-4o's fine-grained annotations, even though many LLM judges are commonly trained on synthetic data [1][2][3]. Indeed, we observed potential biases when using GPT-4o to annotate training data. We have identified the following main types of bias:
> 1. **Annotation bias due to insufficient understanding of evaluation dimensions**. We required GPT-4o to provide not only preference labels but also analysis and justification for its annotations. Through this process, we found that GPT-4o occasionally made inaccurate annotations due to a lack of understanding of the dimensions’ meaning.
> 2. **Annotation bias due to unrecognized key dimensions**. Different dimensions have varying degrees of influence on overall preferences. In some cases, preferences on a few critical dimensions can decisively determine the overall winner. However, GPT-4o occasionally leaned towards the response favored by the majority of dimensions. For example, in a scenario with 10 dimensions, if 7 dimensions favored Response A and 3 favored Response B, GPT-4o might select A as the overall winner, potentially overlooking serious flaws in A within the 3 critical dimensions.
>
> Although these biases cannot be entirely eliminated, we implemented several measures to mitigate them. Specifically, we adopted a data pre-screening approach inspired by STaR [3]. During data construction, we required GPT-4o to produce dimension-level preference analyses, providing reasoning for its decisions. We believe that well-reasoned analysis reflect a model's strengths and weaknesses, helping GPT-4o combine all dimensions to determine the correct overall winner. This process is analogous to how robust mathematical reasoning can lead to accurate conclusions. To enhance data quality, we compared GPT-4o's annotated overall winners with the dataset’s ground truth preference labels, retaining 78% of the original data that matched, thereby improving the dataset's quality at the initial stage.
>
> Finally, we extracted 360 samples from the remaining data to create MD-Eval. Through human verification, we found that the agreement between GPT-4o's dimension-level preference annotations and human labels was 82.49%, which is close to the inter-human agreement of 87.61%. This demonstrates the reliability of the data. Moreover, our experimental results on human-annotated benchmarks, such as FLASK Eval, HHH Alignment, and LLMBar, indicate that even though our training set has not been fully manually verified, it has already achieved competitive performance.
>
> ---
>
> > Could you please provide more information on how you verify the quality of MD-Eval test set (used in Table 5) through human evaluation?
>
> To address your concern regarding the quality of GPT-4o annotations, we detail the process of constructing MD-Eval as follows:
>
> 1. **Data Source**: MD-Eval data was derived from a subset of the data constructed in Sections 3.1–3.3, with 10 samples collected for each scenario, resulting in a total of 360 samples (5,482 dimension-level annotations). It is important to note that while using GPT-4o for data annotation, we additionally generated pairwise comparison analyses for each dimension to serve as references for subsequent human annotators.
> 2. **Human Verification Process**: We enlisted five participants to verify the data: three graduate students and two undergraduate interns with foundational NLP research experience. Specifically, the participants reviewed GPT-4o's analysis for each dimension to identify any obvious biases and then provided validated preference annotations. We ensured that these participants annotated every dimension in each sample. Monthly compensation was provided to these participants as remuneration for their dedicated efforts.
> 3. **Final Annotation Agreement**: The final preference label for each dimension was determined by the majority vote among participants. We defined the inter-annotator agreement as the average proportion of participants selecting the most chosen preference for each dimension. The agreement among the five annotators was calculated to be 87.61%.
> 4. **Comparison with GPT-4o Annotations**: We compared the human-validated labels with GPT-4o annotations and found an agreement of 82.49%.
>
> We will include these quality control details in the appendix.

---

> ### Author Response · Authors · 2024-11-25
>
> We sincerely appreciate the time and effort you have dedicated to reviewing our manuscript and providing valuable suggestions! As the author-reviewer discussion phase draws to a close, we would like to confirm whether our responses have effectively addressed your concerns. A few days ago, we provided detailed replies to your comments, and we hope they have adequately resolved the issues you raised. Should you require further clarification or have any additional questions, please do not hesitate to reach out to us. We are more than willing to continue the communication with you. Thank you once again, and we look forward to your further feedback!

---

> > ### Comment · Reviewer_EYLr · 2024-11-28
> >
> > Thanks for your detailed responses. After carefully checking the additional results, I have raised my score.

---

### Official Review · Reviewer_o3px · 2024-11-06

**Soundness:** 3
**Presentation:** 3
**Contribution:** 3
**Rating:** 8
**Confidence:** 5

**Summary:**

This paper introduces SaMer, a novel scenario-aware multi-dimensional evaluator designed to assess the quality of Large Language Model (LLM) responses to open-ended questions. The work addresses a significant challenge in LLM evaluation: the complex, multifaceted nature of response quality assessment, which varies significantly across different types of queries and use cases. While traditional LLM evaluators typically provide only overall scores or rankings, with some offering limited multidimensional evaluations across fixed dimensions like harmlessness, honesty, and helpfulness, different scenarios require different evaluation criteria. The key innovation of SaMer lies in its ability to provide both overall and fine-grained assessments of LLM-generated responses, with a flexible framework that adaptively identifies and prioritizes relevant evaluation dimensions based on the query type. This approach contrasts with existing methods that rely on predetermined evaluation dimensions. To support this framework, the authors developed a large-scale fine-grained preference dataset encompassing 42 evaluation dimensions across 36 scenarios, with scenarios categorized into five main types based on Maslow's hierarchy. Three graduate students created dimension definitions, and the final dataset contains 135,402 total data points, with 2,000-5,000 samples per scenario.
The dataset construction started with the collection of preference data (win/lose/tie labels) from multiple sources, including custom datasets built using LLM-generated annotations. Scenario annotation utilized a custom-trained LLaMA-3-8B classifier on [Li et al., 2023] data, enriched with GPT4-o-mini annotations for target scenarios without previously existing annotations. The sampling process initially gathered 6,000+ instances per scenario, followed by GPT-4o-mini verification of scenario labels and final balancing to 2,000-5,000 samples per scenario. Fine-grained preference annotation was performed using GPT-4o.
The model architecture consists of a frozen pre-trained LLaMA-3-8B (pre-trained for rewarding using [Wang et al., 2024]) as the text encoder, with its original output projection layer removed. On top of this encoder, the model employs three specialized MLP heads: a quality dimension prediction layer performing multi-label classification using ZLPR loss, a pairwise preference learning layer using margin ranking loss, and a weighting layer for overall scoring also utilizing margin ranking loss. The training process combines these losses (L_{zlpr}, L_{dim}, and L_{o}) through weight hyperparameters lambda.
The evaluation of SaMer was conducted across 9 benchmarks spanning 3 families. For single rating benchmarks, the assessment used Pearson, Spearman, and Kendall-Tau correlations, while preference selection tasks were evaluated using accuracy metrics for pairwise and fine-grained comparisons. The results demonstrate SaMer's superior performance compared to existing baselines, showing strong adaptability across diverse scenarios while providing interpretable, fine-grained evaluations.

**Strengths:**

- The paper is clear and well-structured, effectively communicating complex ideas through a logical flow and professional presentation.
- The fine-grained preference dataset represents a valuable contribution to the field.
- The model is efficient and practical, making it accessible for deployment and fine-tuning.
- The evaluation is thorough and comprehensive, benchmarking against both proprietary models and open-source alternatives.

**Weaknesses:**

- From a methodological standpoint, the paper's novelty is limited. While it combines multiple components to create a comprehensive evaluation system, the individual techniques used are standard approaches in the field, and many data resources are derived. The main innovation lies in the application rather than the methodology itself.
- The hyperparameter analysis lacks depth and transparency. For example, the authors fix lambda values to combine the different losses without exploring or discussing their impact on model performance. Similarly, other hyperparameters are presented with single values, offering no insights into their selection process or their effects on the model's behavior. This absence of ablation studies or sensitivity analyses makes it difficult to understand the robustness of the approach.
- The use of GPT-4o-based models on this scale for intensive annotation and filtering is resource-intensive and appreciable. However, this heavy reliance can introduce potential biases and raises questions about the overall training dataset quality.
- There is no discussion of verification steps or quality control through human analysis to ensure the artificial annotations' reliability. This is particularly important given the subjective nature of many evaluation dimensions.
- The supplementary materials lack data samples, making it impossible for reviewers to assess the quality and characteristics of the dataset.
- The metrics section (5.2) requires a more detailed explanation.
- The paper does not clearly provide information about code availability, model checkpoint distribution, or dataset licensing.

**Questions:**

Please specify the version of the employed proprietary models to ensure reproducibility.

---

> ### Author Response · Authors · 2024-11-21
> **Response to Reviewer o3px (1)**
>
> Thanks for your comments!
> > From a methodological standpoint, the paper's novelty is limited. While it combines multiple components to create a comprehensive evaluation system, the individual techniques used are standard approaches in the field, and many data resources are derived. The main innovation lies in the application rather than the methodology itself.
>
> We acknowledge that our training techniques are based on standard methods in the field of LLM evaluation and our initial data is sourced from existing open-source datasets, we believe that the core innovation and contribution of our work lie in the design of the evaluator. To the best of our knowledge, **SaMer is the first LLM evaluator trained on a multi-dimensional, multi-scenario preference dataset that can adaptively identify the required evaluation dimensions for each query and provide scores at both the dimension-level and instance-level.** Additionally, we plan to release our dataset, contributing a dimension-level preference dataset of 135K examples to the community.
>
> ---
> > The hyperparameter analysis lacks depth and transparency. For example, the authors fix lambda values to combine the different losses without exploring or discussing their impact on model performance. Similarly, other hyperparameters are presented with single values, offering no insights into their selection process or their effects on the model's behavior. This absence of ablation studies or sensitivity analyses makes it difficult to understand the robustness of the approach.
>
> Thank you very much for your valuable suggestions! To further analyze the robustness of the SaMer training method, we investigated the model's performance under different values of λ1 and λ2 in the single rating and pairwise comparison evaluation modes, as shown in **Table R2**. Specifically, we used Vicuna Bench to evaluate the performance in the single rating task and Auto-J Eval for the pairwise comparison task.
>
> In Table R2, we explored the effects of various combinations of λ1 and λ2 values on SaMer's performance by increasing or decreasing their values. When λ1 > λ2 (e.g., λ1 = 1.5, λ2 = 1), we observed an improvement in SaMer's performance on the pairwise comparison task, while its performance in the single rating task slightly declined. We attribute this to the fact that, in the pairwise comparison setting, SaMer evaluates two responses using the same evaluation dimensions and their associated weights, making dimension-level scoring particularly important. Since λ1 governs the dimension-level rank loss, it directly impacts this task. However, in the single rating setting, accurately predicting evaluation dimensions and their weights becomes more critical, and too low a value of λ2 hinders effective weight prediction.
>
> When λ1 < λ2 (e.g., λ1 = 1, λ2 = 1.5), we observed improved performance in the single rating task but reduced performance in the pairwise comparison task. This indicates that increasing λ2 indeed aids in predicting dimension weights, thereby improving the accuracy of single ratings, further validating the earlier analysis. However, we also noticed that under the configuration λ1 = 0.5 and λ2 = 1, both single rating and pairwise comparison performance deteriorated, emphasizing the importance of dimension-level scoring (i.e., λ1).
>
> In summary, considering the respective roles and significance of λ1 and λ2, the design choice of setting λ1 = λ2 = 1 is both reasonable and effective.
>
> **Table R2**. Impact of λ1 and λ2 on SaMer Performance in Single Rating and Pairwise Comparison Tasks
> |λ1,λ2|Vicuna Bench|Auto-J Eval (w/o tie)|Auto-J Eval (w/ tie)|
> |-----|------------|---------------------|--------------------|
> |1,1(default)|47.59|76.15|57.61|
> |0.5,1|47.15|75.86|57.33|
> |1.5,1|47.58|76.25|57.76|
> |1,0.5|47.39|76.35|57.76|
> |1,1.5|47.97|75.37|56.97|

---

> ### Author Response · Authors · 2024-11-21
> **Response to Reviewer o3px (2)**
>
> Thanks for your valuable comments!
> > The use of GPT-4o-based models on this scale for intensive annotation and filtering is resource-intensive and appreciable. However, this heavy reliance can introduce potential biases and raises questions about the overall training dataset quality.
>
> Your concern about the quality of the training data is well-founded, even though many LLM judges are commonly trained on synthetic data [1][2][3]. Indeed, we observed potential biases when using GPT-4o to annotate training data. We have identified the following main types of bias:
>
> 1. **Annotation bias due to insufficient understanding of evaluation dimensions**. We required GPT-4o to provide not only preference labels but also analysis and justification for its annotations. Through this process, we found that GPT-4o occasionally made inaccurate annotations due to a lack of understanding of the dimensions’ meaning.
>
> 2. **Annotation bias due to unrecognized key dimensions**. Different dimensions have varying degrees of influence on overall preferences. In some cases, preferences on a few critical dimensions can decisively determine the overall winner. However, GPT-4o occasionally leaned towards the response favored by the majority of dimensions. For example, in a scenario with 10 dimensions, if 7 dimensions favored Response A and 3 favored Response B, GPT-4o might select A as the overall winner, potentially overlooking serious flaws in A within the 3 critical dimensions.
>
> Although these biases cannot be entirely eliminated, we implemented several measures to mitigate them. Specifically, we adopted a data filtering approach inspired by STaR [4]. During data synthesis, we first asked GPT-4o to perform fine-grained analysis and preference annotation for individual dimensions. Finally, we required GPT-4o to synthesize these dimension-level preferences to decide the overall winner. Ideally, accurate dimension-level evaluations provide a thorough analysis of a response's strengths or critical flaws, which in turn influence the final decision for the overall winner. We liken this task to the intermediate reasoning process in STaR for mathematical problems, where a sound reasoning process often leads to a correct final answer.
>
> Using pre-collected open-source datasets that include human-annotated winners, we filtered out unreliable annotations by comparing GPT-4o's overall winner annotations with human annotations. Through this filtering process, we retained 78% of the original data.
> Finally, although the training data was generated by GPT-4o, we extracted a portion of the data for manual verification and created a test set (MD-Eval). In this process, we observed an agreement of 82.49% between GPT-4o's annotations and human labels, compared to an inter-human agreement of 87.61%.
>
> [1] Kim S, Shin J, Cho Y, et al. Prometheus: Inducing fine-grained evaluation capability in language models. ICLR2024.
>
> [2] Li J, Sun S, Yuan W, et al. Generative Judge for Evaluating Alignment. ICLR2024.
>
> [3] Park J, Jwa S, Meiying R, et al. OffsetBias: Leveraging Debiased Data for Tuning Evaluators[C]. EMNLP 2024.
>
> [4] Zelikman E, Wu Y, Mu J, et al. Star: Bootstrapping reasoning with reasoning. NIPS 2023.
>
> ---
> > There is no discussion of verification steps or quality control through human analysis to ensure the artificial annotations' reliability. This is particularly important given the subjective nature of many evaluation dimensions.
>
> To address your concern regarding the quality of GPT-4o annotations, we detail the process of constructing MD-Eval as follows:
>
> 1. Data Source: MD-Eval data was derived from a subset of the data constructed in Sections 3.1–3.3, with 10 samples collected for each scenario, resulting in a total of 360 samples (5,482 dimension-level annotations). It is important to note that while using GPT-4o for data annotation, we additionally generated pairwise comparison analyses for each dimension to serve as references for subsequent human annotators.
>
> 2. Human Verification Process: We enlisted three graduate students with foundational NLP research experience to verify the data. Specifically, the participants reviewed GPT-4o's analysis for each dimension to identify any obvious biases and then provided validated preference annotations. We ensured that these participants annotated every dimension in each sample.
>
> 3. Final Annotation Agreement: The final preference label for each dimension was determined by the majority vote among participants. We defined the inter-annotator agreement as the average proportion of participants selecting the most chosen preference for each dimension. The agreement among the five annotators was calculated to be 87.61%.
>
> 4. Comparison with GPT-4o Annotations: We compared the human-validated labels with GPT-4o annotations and found an agreement of 82.49%.
>
> We will include these quality control details in the appendix.

---

> ### Author Response · Authors · 2024-11-21
> **Response to Reviewer o3px (3)**
>
> Thank you for your suggestion!
> > The supplementary materials lack data samples, making it impossible for reviewers to assess the quality and characteristics of the dataset.
>
> We have added some data samples, including a portion of the training data and the complete MD-Eval dataset. Please refer to the following link for details: https://anonymous.4open.science/r/SaMer-2878/README.md.
>
> ---
>
> > The metrics section (5.2) requires a more detailed explanation.
>
> Thank you for pointing out this improvement opportunity. We have added more details about the metrics in Section 5.2.
> For the Single Rating setup, we use three different correlation metrics—Pearson, Spearman, and Kendall-Tau—to evaluate the correlation between the model-generated scores and the reference scores.
> For the Pairwise Comparison and Fine-Grained Comparison (Multi-dimensional) setups, we assess whether the preferences predicted by the model align with the reference labels. Accordingly, we use accuracy as the evaluation metric.
>
> ---
>
> > The paper does not clearly provide information about code availability, model checkpoint distribution, or dataset licensing.
>
> Thank you for your feedback! We have made the training and inference code, model checkpoints, and dataset license publicly available. Please refer to the following link for details: https://anonymous.4open.science/r/SaMer-2878/README.md.

---

> ### Author Response · Authors · 2024-11-25
>
> We sincerely appreciate the time and effort you have dedicated to reviewing our manuscript and providing valuable suggestions. As the author-reviewer discussion phase draws to a close, we would like to confirm whether our responses have effectively addressed your concerns. A few days ago, we provided detailed replies to your comments, and we hope they have adequately resolved the issues you raised. Should you require further clarification or have any additional questions, please do not hesitate to reach out to us. We are more than willing to continue the communication with you. Thank you once again, and we look forward to your further feedback!

---

> > ### Comment · Reviewer_o3px · 2024-11-25
> >
> > Thank you for the exhaustive reply. Based on the additional comments, I have increased my score.

---

### Author Response · Authors · 2024-11-22
**General Response to Reviewers and Revision Submitted**

We sincerely thank all reviewers for their insightful comments and suggestions! While we reply to the comments of each reviewer separately, we summarize below the major revisions made to address the reviewers' concerns.

The main revisions are as follows:

1. In Appendix A1, we have added specific definitions of evaluation dimensions for certain scenarios. (Reviewers EYLr, DeNC)

2. In Appendix A2, we have included details about data annotation, including the prompts used for fine-grained annotation with GPT-4o and the initial data filtering process. (Reviewers o3px, EYLr, DeNC)

3. In Appendix A3, we have provided a detailed quality control process, including scenario-specific evaluation dimension decisions in A3.1 and manual validation of MD-Eval in A3.2. (Reviewers o3px, EYLr, DeNC)

---

### Meta-Review · Area_Chair_YCFt · 2024-12-18

**Metareview:**

This paper proposes a scenario-aware multi-dimensional evaluator designed to provide both overall and fine-grained assessments of LLM-generated responses. Experiments on eight single rating and pairwise comparison datasets demonstrate that SaMer outperforms existing evaluators in a variety of evaluation tasks. The proposed evaluator is well motivated and the results are promising. The paper is clear and well-structured. The fine-grained preference dataset may be a valuable resource for future research. The reviewers raised a few issues about the presentation and evaluation, and were generally satisfied with the authors' rebuttal. All the reviewer were ultimately positive about this paper.

**Additional Comments On Reviewer Discussion:**

The reviewers were generally satisfied with the authors' rebuttal and they increased there scores during the rebuttal period.

---

### Decision · Program_Chairs · 2025-01-22

Accept (Poster)